# Elevated DNA damage without signs of aging in the short-sleeping Mexican cavefish

Evan Lloyd[1†], Fanning Xia[2†], Kinsley Moore[1], Carolina Zertuche Mery[1], Aakriti Rastogi[1], Robert A Kozol[3], Olga Kenzor[2], Wesley Warren[4], Lior Appelbaum[5], Rachel L Moran[1], Chongbei Zhao[2], Erik R Duboue[3], Nicolas Rohner[2]*, Alex C Keene[1]*

[1]Department of Biology, Texas A&M University, College Station, United States; [2]Stowers Institute for Medical Research, Kansas City, United States; [3]Harriet Wilkes Honors College, Florida Atlantic University, Jupiter, United States; [4]Department of Genomics, University of Missouri, Columbia, United States; [5]Faculty of Life Science and the Multidisciplinary Brain Research Center, Bar Illan University, Ramat Gan, Israel

*For correspondence:
nro@stowers.org (NR);
AlexCKeene@gmail.com (ACK)

†These authors contributed equally to this work

Competing interest: The authors declare that no competing interests exist.

## eLife Assessment

Lloyd et al. used an evolutionary comparative approach to study DNA damage repair associated with low sleep duration in Astyanax mexicanus, highlighting how the cavefish population has evolved a reduced DNA damage response. The results presented here have **important** implications. Their results are generally **solid**, however the evidence suggesting that sleep differences are linked to DNA damage response is missing and this hypothesis remains to be fully tested.

**Abstract** Dysregulated sleep has widespread health consequences, including the accumulation of DNA damage. The Mexican tetra, *Astyanax mexicanus*, provides a powerful model to study the evolution and consequences of sleep loss. Multiple cave-adapted populations of this species have independently evolved reduced sleep compared to surface populations, yet show no obvious decline in healthspan or longevity. To examine whether evolved sleep loss is associated with DNA damage, we compared DNA damage response (DDR) and oxidative stress across populations. Cavefish exhibited elevated γH2AX in the brain and increased gut oxidative stress, consistent with chronic sleep deprivation. Following acute UV exposure, surface fish, but not cavefish, increased sleep and activated the photoreactivation repair pathway. Fibroblast cell lines derived from both populations confirmed diminished DDR and repair in cavefish, supporting an attenuated acute DNA damage response. Transcriptomic analysis revealed that many genes differentially expressed with aging in surface fish remain unchanged in cavefish, suggesting altered regulation of aging-related pathways. Together, these findings indicate that cavefish experience elevated cellular hallmarks of sleep deprivation yet exhibit resilience to its long-term consequences, highlighting an evolutionary model to investigate the mechanisms underlying sleep, DNA repair, and healthy aging.

## Introduction

Sleep is ubiquitous throughout the animal kingdom and has been identified in animals with relatively simple neural networks, including jellyfish and nematodes, through primates, suggesting ancient function and shared evolutionary origins (*Aulsebrook et al., 2016*; *Joiner, 2016*; *Keene and Duboue, 2018*; *McNamara et al., 2009*; *Zimmerman et al., 2008*). While the primary functions

of sleep are not fully understood, it is essential for many processes, including neural connectivity, clearance of toxic metabolites, immunity, learning, and memory (*Frank, 2020*; *Hartmann, 1973*; *Siegel, 2005*). There is growing evidence that DNA damage may play an important role in sleep drive (*Carroll et al., 2016*; *Vaccaro et al., 2020*; *Zada et al., 2021*; *Zada et al., 2019*). DNA damage is associated with periods of prolonged wakefulness and is reduced during sleep across numerous species, including *C. elegans*, zebrafish, mice, and humans (*Bellesi et al., 2016*; *Cheung et al., 2019*; *Goetting et al., 2020*). In turn, sleep disruption is associated with DNA damage, and sleep deprivation (SD) inhibits the expression of DNA repair genes in humans (*Carroll et al., 2016*; *Goetting et al., 2020*), suggesting a critical role for sleep in the maintenance of genome integrity and function and an association between sleep loss and DNA damage, which could lead to neurodegeneration. Furthermore, sleep loss results in elevated reactive oxygen species (ROS), a known mediator of DNA damage, in the gut and/or brain that contribute to mortality in *Drosophila* (*Haynes et al., 2024*; *Vaccaro et al., 2020*). The effects of sleep loss on ROS production in the gut is also present in mice (*Vaccaro et al., 2020*). Despite these advances, little is known about the cellular consequences of sleep loss or the evolutionary relationship between DNA damage and sleep regulation.

Comparative approaches examining evolutionarily derived differences in sleep have provided significant insight into the genetic and functional basis of sleep regulation (*Allada and Siegel, 2008*; *Cirelli, 2009*; *Zimmerman et al., 2008*). While the majority of sleep studies in fish have used zebrafish, the Mexican tetra *Astyanax mexicanus* is an emerging model for investigating the genetic and evolutionary basis underlying behavioral and physiological traits (*Keene et al., 2015*; *Keene and Appelbaum, 2019*; *Kowalko, 2020*; *McGaugh et al., 2020a*; *Yoshizawa, 2015*). *A. mexicanus* exists as blind cave populations and an extant surface population; while the surface and cave populations are geographically isolated, they remain interfertile and capable of hybridization in natural and laboratory settings (*Moran et al., 2023*). In this system, there are similar sleep loss phenotypes among geographically and geologically isolated cave populations (*Aspiras et al., 2015*; *Duboué et al., 2011*) with likely unique genetic bases between caves (*Duboué et al., 2011*; *Mack et al., 2021*; *Yoshizawa et al., 2015*). Furthermore, evolved differences in DNA repair genes have been identified across all three cave populations studied to date, including links between mechanisms regulating sleep, light responsiveness, and DNA repair (*Beale et al., 2013*; *Mack et al., 2021*). These findings support the notion that the genetic and molecular underpinnings of sleep are closely related to DNA repair processes in cavefish.

Examining the ecological factors that contribute to evolved changes in sleep regulation and the physiological consequences of this sleep loss has the potential to address the fundamental functions of sleep. For example, in humans, insomnia is associated with many different diseases and increased morbidity, suggesting that sleep is critical for healthy aging (*Arble et al., 2015*; *Carroll and Prather, 2021*). Strikingly, cavefish sleep as little as 1–2 hr per day, in contrast to their surface counterparts, which sleep as much as 6–10 hr a day (*Duboué et al., 2011*). Despite the dramatic reduction in sleep, there are no apparent health consequences to cavefish, suggesting an established resilience to sleep loss (*Cobham and Rohner, 2024*; *Riddle et al., 2018a*; *Rohner, 2018*). Considering that changes in the levels of DNA breaks are associated with sleep regulation in flies, zebrafish, mice, and humans (*Bellesi et al., 2016*; *Carroll et al., 2016*; *Zada et al., 2021*), it is possible that intrinsic changes in the DNA repair and DDR pathways underlie reduced sleep need in cavefish.

Here, we sought to investigate the relationship between DNA damage and the evolution of sleep, to test at the cellular and organismal levels whether cavefish have markers of chronic sleep deprivation and accelerated aging. We find that DNA damage is elevated in cavefish brains, consistent with the notion that sleep loss is associated with elevated levels of DNA damage. The transcriptional and behavioral response to UV damage is blunted in cavefish, and cavefish cells exhibit diminished DNA repair capabilities, raising the possibility that reduced DDR function may contribute to sleep loss in cavefish. To examine the long-term consequences of reduced DDR, we examined transcriptional differences in young and aged surface and cavefish. While aging in surface fish is associated with broad transcriptional changes across tissue types, there are relatively few transcriptional differences between young and aged cavefish. Together, these findings reveal that cavefish appear to have developed molecular resilience to aging despite elevated DNA damage that likely derives from sleep loss.

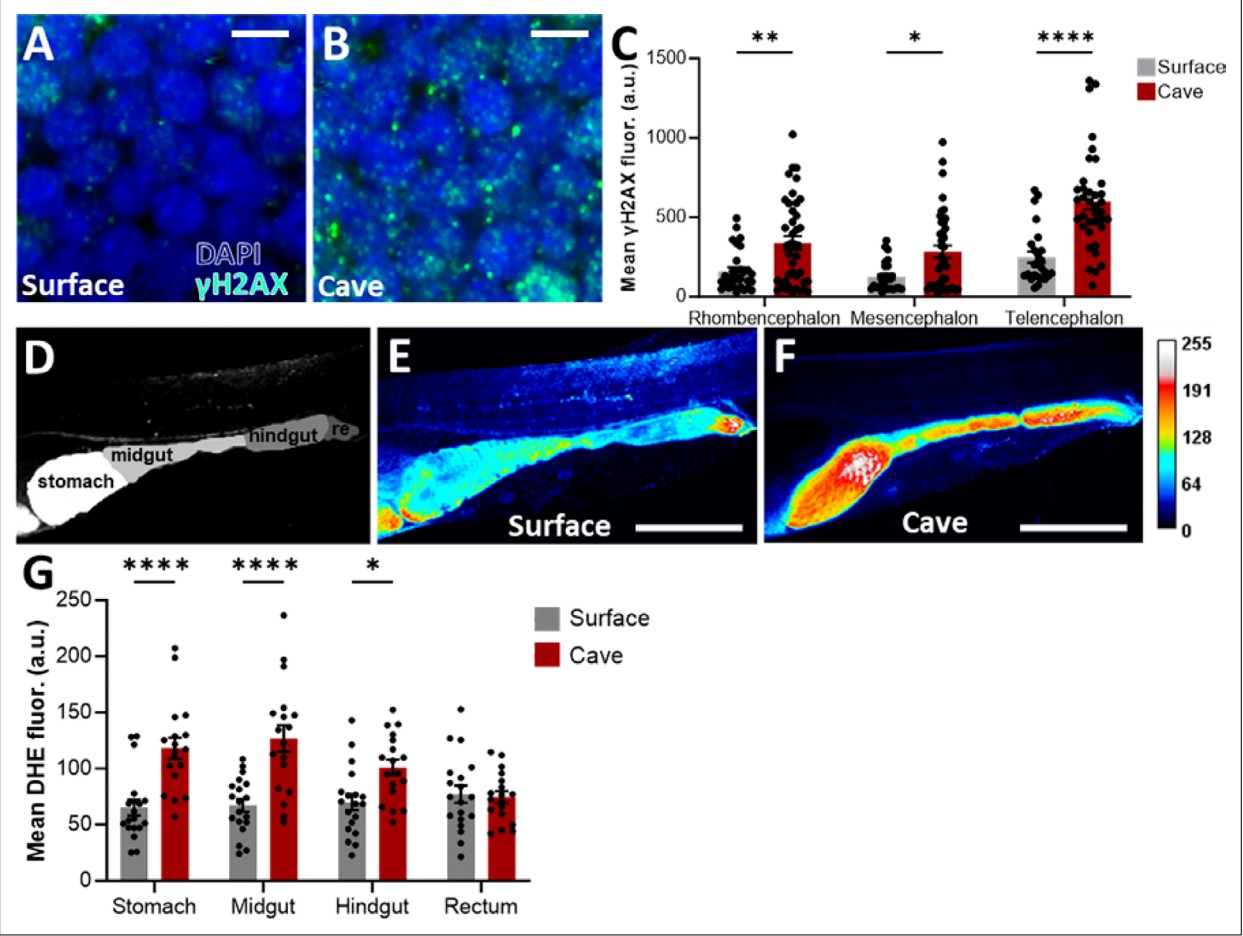

**Figure 1.** Cavefish harbor increased neuronal DNA damage and gut reactive oxygen species (ROS). (**A, B**) Representative images of cells stained with DAPI and γH2AX in the rhombencephalon of surface fish (**A**) and cavefish (**B**). Scale bar = 5 µm. (**C**) Mean γH2AX fluorescence across three regions of surface fish and cavefish brains. (rhomb: rhombencephalon; mes: mesencephalon; tele: telencephalon) (Mixed-effects analysis: $F_{1, 68} = 32.08$, $p < 0.0001$; surface n = 29, cave n = 41). (**D**) Representative image of larval gut showing regions in false color (re: rectum). (**E, F**) Representative images of surface fish and cavefish guts stained with dihydroethidium (DHE) marking ROS. Scale bar = 500 µm. (**G**) Mean DHE fluorescence across four regions of surface fish and cavefish guts (two-way repeated measures ANOVA: $F_{1, 35} = 48.36$, $p < 0.0001$, surface n = 19, cave n = 20). All error bars represent S.E.M.

The online version of this article includes the following video for figure 1:

**Figure 1—video 1.** DNA damage in the cavefish brain.

https://elifesciences.org/articles/99191/figures#fig1video1

## Results

In vertebrates, including teleost fish such as zebrafish and *A. mexicanus*, DNA damage activates a highly conserved and stereotypical response and repair program. Markers of DDR proteins can be localized and quantified at the cellular and whole-organism levels. Phosphorylation of histone H2AX on serine 139 (γH2AX) offers a well-established and quantifiable biomarker of cellular response to DNA double-strand breaks that is necessary for the assembly of repair complexes (*Siddiqui et al., 2015*). We compared γH2AX between surface fish and Pachón cavefish brains at ZT0, a time when both populations are behaviorally active and sleep is minimal (*Figure 1A–B* and *Figure 1—video 1*). Levels of γH2AX were elevated in the rhombencephalon, telencephalon, and mesencephalon of cavefish compared to surface fish (*Figure 1C*). These findings suggest that DNA damage is elevated in short-sleeping cavefish compared to their surface fish counterparts.

Sleep loss is associated with reduced gut function, including the accumulation of ROS (*Vaccaro et al., 2020*). To examine whether markers of sleep loss extend to the gut in *A. mexicanus*, we compared ROS in the guts of surface and cavefish. Fish aged 6 days post-fertilization (dpf) were incubated in the ROS marker dihydroethidium (DHE), and guts were imaged on a confocal microscope.

We found that ROS in the gut is upregulated in cavefish, reinforcing the idea that cellular stress and canonical markers of DDR are elevated in these short-sleeping fish (*Figure 1D–G*); ROS levels were elevated in the stomach, midgut, and hindgut, but not in the rectum (*Figure 1G*). Prior to imaging, both surface and cavefish had been reared in a temperature-controlled incubator and relied solely on their yolk sac for nutrients; so, differences in gut ROS cannot be attributed to differences in rearing or feeding conditions. Together, these findings fortify the notion that cellular stress is elevated in the gut of cavefish relative to surface fish.

To further examine the direct link between DNA damage and sleep in *A. mexicanus*, surface fish and cavefish were exposed to short periods of UV-B radiation, known to cause DNA damage and induce double-stranded breaks (*Zada et al., 2021*). In surface fish, UV exposure resulted in a dose-dependent increase in sleep for up to 3 hr, similar to findings in zebrafish (*Zada et al., 2021*; *Figure 2A and B*). Interestingly, sleep decreased in surface fish sleep during the nighttime, perhaps the result of sleep credit that derives from increased sleep during the day (*Figure 2A*; *Öztürk-Çolak et al., 2020*). Increases in sleep amount in surface fish were mediated by both increased bout number, and, at the higher dose, an increase in bout length (*Figure 2C and D*). Conversely, there was no effect of UV treatment on daytime or nighttime sleep in cavefish (*Figure 2E–H*).

Analysis of sleep- and wake-probability was consistent with these measurements, showing elevated sleep pressure and reduced wake pressure in surface fish during the 3 hr following treatment; in cavefish, there was no change (*Figure 2—figure supplement 1*). These results confirm that UV-induced DNA damage promotes sleep in *A. mexicanus* surface fish, whereas this response is lost in cavefish.

We examined the effects of the high dose (60 s) of UV treatment at zeitgeber time (ZT) 0 (onset of lights on) on the transcriptional response in surface fish and cavefish. Fish aged 6 dpf were harvested for RNA extraction 90 min following UV exposure (*Figure 3—figure supplement 1*). PCA analysis of RNA expression showed that the largest factor driving variability in transcriptional response across samples was population, with principal component 1 separating samples by population and accounting for 86% of the variance. Principal component 2 separated samples by treatment and accounted for 9% of the variance (*Figure 3B*). Analysis revealed numerous genes that were differentially expressed in both populations, including upregulation of the RNA Polymerase regulating transcription factor *fosl1a* and downregulation of *spi-c* (*Figure 3C*). Numerous genes were selectively differentially expressed, including upregulation of the heat shock protein *hspb9* in surface fish and downregulation of the glucose sensor *gck* in cavefish (*Figure 3C*, *Figure 3—figure supplement 1*). To determine if the DDR pathway is activated in cavefish following exposure to UV light, we quantified changes in pathway components in surface fish and cavefish. A heat map of DNA repair genes significantly upregulated in surface fish revealed that several components of DNA repair pathways are differentially expressed in cavefish following UV treatment (*Figure 3D*). Notably, cyclobutane pyrimidine dimer photolyase (*cpdp*), an important component of the photoreactivation repair pathway for UV-induced DNA damage, responds strongly in surface fish, but fails to respond in cavefish. Although previous groups have reported that *cpdp* is constitutively upregulated in adult cavefish (*Beale et al., 2013*), our analysis did not align with this finding; however, this could be due to differences in the age or circadian time point studied. A more detailed analysis of previous circadian transcriptomic studies in cavefish revealed that two of the DDR genes are elevated at some, but not all, phases of the circadian cycle (*Figure 3—figure supplement 2*). Additionally, *xrcc3*, a paralog of human *rad51* that is important for homologous recombination, and *fan1*, a component of the Fanconi Anemia pathway, are nonresponsive to UV-B treatment in cavefish. Conversely, *ube2al*, which is required for post-replication DNA repair, is perhaps even more strongly activated in cavefish compared to surface fish. Together, these findings suggest that DNA repair processes in cavefish have undergone complex changes, with some pathways rendered nonfunctional, while others may have been upregulated to compensate for the loss of function in other areas. To understand transcriptional changes more fully as a result of UV-B treatment, gene set enrichment analysis (GSEA) was performed on both surface fish and cavefish. A large number of pathways were enriched in both populations (*Figure 3—figure supplement 1*); to examine differences between the transcriptional responses of the two populations, we identified pathways which were specifically enriched in either surface fish or cavefish (*Figure 3E*). Only surface fish showed significant activation of genes associated with response to ROS and cell redox homeostasis, consistent with measurements of elevated ROS in cavefish, whereas cavefish showed activation of genes

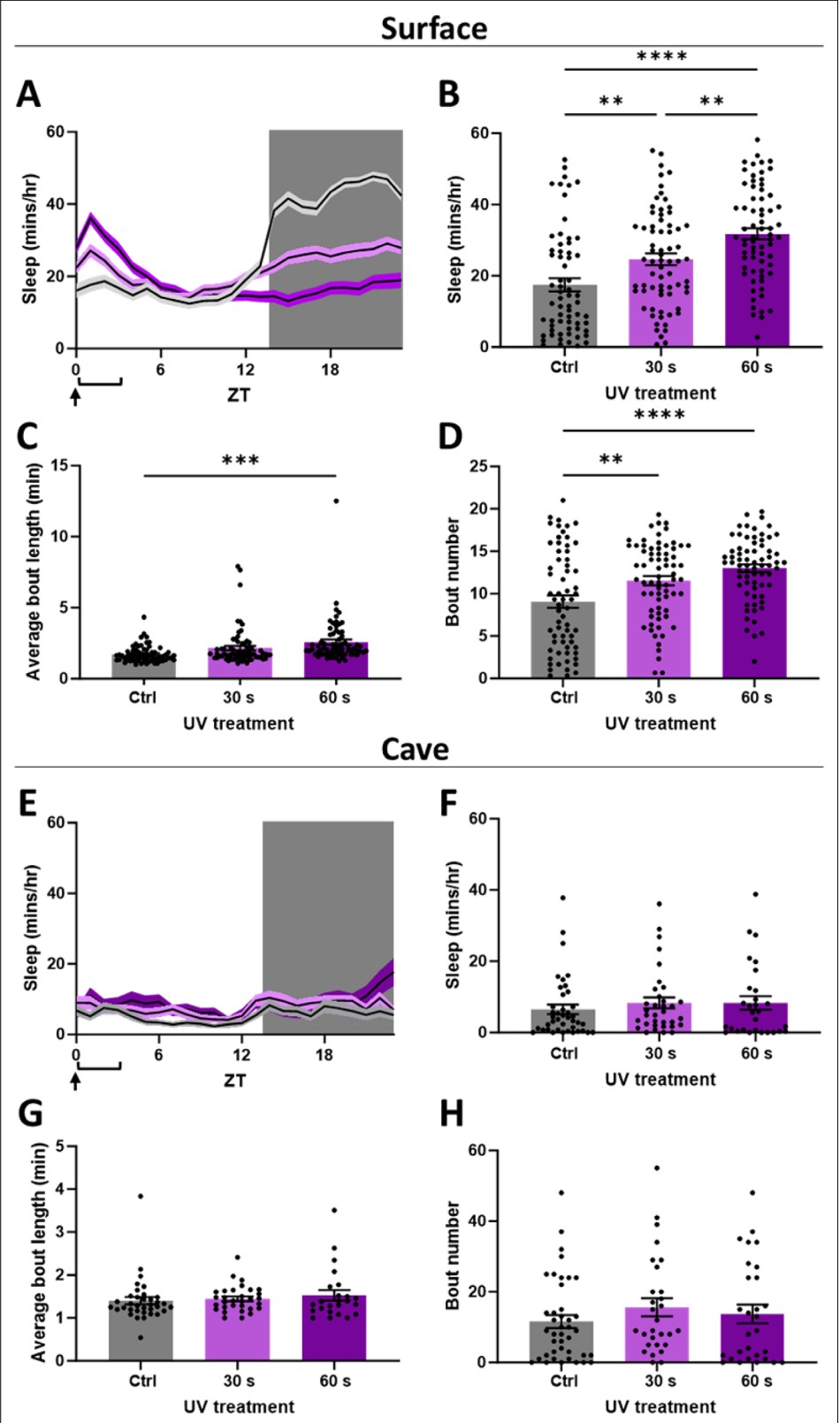

**Figure 2.** Cavefish lack a sleep response to UV-B-induced DNA damage. (**A**) The 24 hr sleep profiles of surface fish exposed to 30 or 60 s of UV-B light compared to controls. (**B**) Average sleep amount in surface fish in the 3 hr following UV-B exposure (one-way ANOVA: $F2$, 202 = 18.75, $p < 0.0001$). (**C**) Average bout length in surface fish in the 3 hr following UV-B exposure (one-way ANOVA: $F2$, 201 = 8.301, $p = 0.0003$). (**D**) Bout number in the

*Figure 2 continued on next page*

*Figure 2 continued*

3 hr following UV-B exposure (one-way ANOVA: $F_2$, 201 = 11.5, $p$<0.0001). (**E**) The 24 hr sleep profiles of cavefish exposed to 30 or 60 s of UV-B light compared to controls. (**F**) Average sleep amount in cavefish in the 3 hr following UV-B exposure. (**G**) Average bout length in cavefish in the 3 hr following UV-B exposure. (**H**) Bout the number of cavefish in the 3 hr following UV-B exposure. (ZT = Zeitgeber time). All treatments performed at ZT0. Surface: Ctrl n = 65, 30s n = 70, 60s n = 70; Cave: Ctrl n = 41, 30s n = 34, 60s n = 29. All error bars represent S.E.M.

The online version of this article includes the following figure supplement(s) for figure 2:

**Figure supplement 1.** Response in sleep pressure to UV-B-induced DNA damage.)

associated with metabolic processes and suppression of genes associated with synaptic signaling, suggesting altered responses to DNA damage.

To quantify DNA damage on a cellular level, we established embryonic fibroblast cell lines derived from surface fish and Pachón cavefish embryos (*Figure 4—figure supplement 1*). In brief, we dissociated and sterilized cells from embryos less than 12 hr post-fertilization and isolated individual clones that were propagated for over 40 passages. To confirm that the cell lines indeed represented fibroblasts, we stained the cells for the presence of vimentin, a known fibroblast marker (*Satelli and Li, 2011*). Both cell lines exhibited stable vimentin expression (*Figure 4—figure supplement 1*). We further validated the fibroblast nature of the cell lines by RNA sequencing, which showed enhanced expression of other fibroblast signature genes such as *col1a1* compared to other cell types, including the previously established liver-derived cell lines (*Krishnan et al., 2022*) from the same species, mouse stem cells, and mouse embryonic fibroblasts (*Wang et al., 2019*; *Figure 4—figure supplement 1*).

We next used the newly generated fibroblast cell lines to measure the DNA damage level upon UV radiation exposure. We exposed the cells to 100 J/m$^2$ UV radiation and visualized UV-induced DNA lesions using an antibody targeting cyclobutene pyrimidine dimer (CPD) as a DNA damage marker. Both the surface fish and cavefish-derived cell lines exhibited strong CPD induction, indicating pronounced UV-induced DNA damage (*Figure 4A*). We quantified the mean fluorescence per nucleus area using the Cellpose function (*Stringer et al., 2021*) of ImageJ and found no significant difference between the two derived cell lines ($p$=0.6404, two-way ANOVA, *Figure 4B*).

To assess whether a similar level of DNA damage in the different cell lines results in variations in DNA damage repair, we quantified γH2AX levels after UV treatment. Measurements were taken bi-hourly for 6 hr post-exposure. The surface fish cells demonstrated a marked increase in γH2AX, whereas the cavefish cell lines showed only a modest rise, implying a diminished UV damage repair capability in cavefish cells (*Figure 4C and D*). To validate these results, we repeated these experiments on liver-derived cell lines (*Krishnan et al., 2022*) and observed a similar trend of reduced levels of γH2AX after radiation exposure in cavefish-derived cells, while the surface fish cells showed strong induction (*Figure 4—figure supplement 1*). This pattern suggests that the DDR differences are not confined to tissue type. To directly test the ability to repair DNA, we employed a host cell reactivation assay (*Nagel et al., 2014*; *Tian et al., 2019*). Briefly, the green fluorescent protein (GFP) plasmid was treated with 600 J/m$^2$ UV. We transfected the cell lines with either an intact or the in vitro UV-damaged GFP plasmid and tracked GFP fluorescence recovery after 50 hr using flow cytometry as a proxy for the ability of the host cell to repair the plasmid (*Figure 4G*). Consistent with the γH2AX findings, the UV-damaged plasmid-transfected cavefish cells displayed a substantially lower GFP signal recovery (~22% relative to control) than the surface fish cells (~49% relative to control) (*Figure 4E and F*). These observations, indicating a diminished DNA repair capacity, align with the sleep behavior differences we observed in UV-treated fish larvae.

DNA damage in the brain, ROS in the gut, and sleep loss are associated with aging (*Carroll and Prather, 2021*; *Maynard et al., 2015*; *Schumacher et al., 2021*). While *A. mexicanus* cavefish have evolved many traits that would be detrimental to humans or other species, there is evidence that they have also evolved metabolic and physiological resilience, enabling them to enjoy a similar or even extended lifespan compared to surface fish (*Lunghi and Bilandžija, 2022*; *Medley et al., 2022*; *Riddle et al., 2018a*). To examine the effects of long-term accumulation of DNA damage in the brain, elevated gut ROS, and sleep loss, we examined the transcriptional profiles of tissue in young and aged fish. Briefly, we dissected the brain, gut, liver, muscle, and heart of young (~1 year old) and aged (7–8 year old) surface fish and Pachón cavefish. PCA analysis of brain samples revealed a strong contribution of population to the sample variance, with samples separated by population across PC1

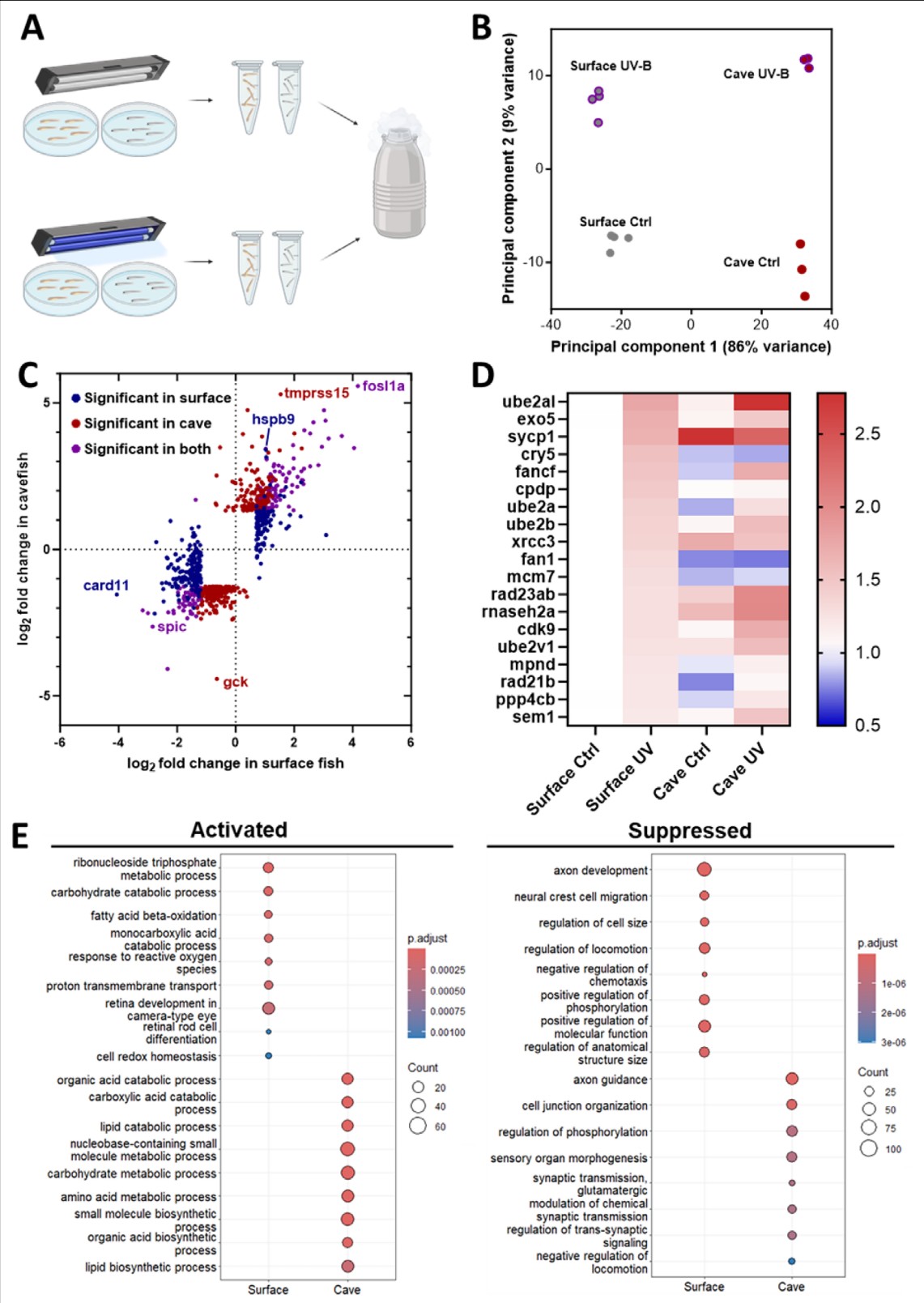

**Figure 3.** Transcriptional responses to UV-B-induced DNA damage in surface fish and cavefish. (**A**) Schematic of experimental design. (**B**) Multidimensional scaling plot depicting the variances in principal component space between the processed sequencing samples. PC1 two-way ANOVA: (Treatment) $F_{1,10}$=6.388, p=0.03, (Population) $F_{1,10}$=4970, p<0.0001, (Interaction) $F_{1,10}$=17.56, p=0.0019. PC2 two-way ANOVA: (Treatment) $F_{1,10}$=465.0, p<0.0001, (Population) $F_{1,10}$=0.4969, p=0.497, (Interaction) $F_{1,10}$=18.56, p=0.0015 (**C**) Bi-directional volcano plot of changes in gene expression in surface

*Figure 3 continued on next page*

*Figure 3 continued*

and cave larvae after exposure to DNA-damaging UV-B radiation. (**D**) Heat map of gene expression in DNA repair genes which responded significantly in UV-B-exposed surface fish.

The online version of this article includes the following figure supplement(s) for figure 3:

**Figure supplement 1.** Transcriptional responses to UV-B-induced DNA damage in surface fish and cavefish.

**Figure supplement 2.** DNA repair gene expression in surface fish and cavefish.

and accounting for 51% of the variance (*Figure 5A*). Interestingly, while surface fish samples were separated by age across PC2, there was no separation of samples by age in cavefish (*Figure 5A*). The same trend held true for gene expression in the gut (*Figure 5—figure supplement 1*). PCA plots of gene expression in the heart and liver did not show clear separation across either population or age, while muscle tissue showed separation by population, but not age (*Figure 5—figure supplement 1*). To examine the impacts on the broader transcriptome, we compared the number of differentially expressed genes between young and aged populations of surface fish and cavefish. Across all tissues, there were markedly more transcripts that were differentially expressed between young and aged surface fish than cavefish in the brain, gut, heart, liver, and muscle (*Figure 5B*). Together, these findings reveal that the transcriptome of cavefish is resilient to age-associated changes despite sleep loss, elevated ROS, and elevated DNA damage.

We sought to examine the specific genes that were differentially expressed between surface fish and cavefish, providing potential mechanisms of resilience to DNA damage and sleep loss. Because sleep is considered necessary, specifically for the repair of neuronal DNA damage, we first examined transcriptional differences in the aging surface fish and cavefish brains. Within the brain, there were only five genes which showed significant changes in both surface fish and cavefish; among these was the gene *top2a*, which is reduced in both populations (*Figure 5C*). *Top2a* is considered essential for structural maintenance of chromosomes during cell division. Intriguingly, GSEA analysis did not reveal any significantly enriched pathways in aged surface fish brains despite the high number of differentially regulated genes, while the aged cavefish brains showed suppression of gene sets related to chromosome condensation and segregation (*Figure 5D*). These processes are known to deteriorate with age because of unrepaired DNA damage, particularly in the brain (*Barroso-Vilares et al., 2020*). Across non-brain tissues, we found enrichment for a wide variety of processes, some of which overlapped across tissues and populations and some of which did not, consistent with the idea that aging is a complex process governed by many factors (*Figure 5—figure supplement 2*). Taken together, these results indicate that, despite elevated levels of DNA damage and impaired DNA damage and repair mechanisms, cavefish are at least partially protected from their harmful effects and exhibit reduced transcriptional changes during aging.

## Discussion

We have investigated differences in DNA damage and the DDR pathway in *A. mexicanus,* a model for evolved sleep loss. UV and other agents that induce DNA damage promote sleep in diverse animals, suggesting a fundamental and highly conserved relationship between DNA damage and sleep regulation (*Bellesi et al., 2016*; *DeBardeleben et al., 2017*; *Zada et al., 2021*). Similarly, we find that DNA damage in the brain and ROS levels in the gut are elevated in Pachón cavefish compared to surface fish. These findings are consistent with the phenotypes of sleep-deprived invertebrates and mammals, supporting the notion that cavefish are sleep-deprived (*Bellesi et al., 2016*; *Vaccaro et al., 2020*). Beyond the Pachón cavefish population, the two other larval cavefish populations tested to date have reduced sleep compared to surface fish (*Duboué et al., 2011*; *Jaggard et al., 2020*). Further investigation of DNA damage in these populations is necessary to determine whether the cellular effects of sleep loss are conserved in independently evolved cavefish populations (*Moran et al., 2022*; *Yoshizawa et al., 2015*). There are many species that have evolved sleep loss, particularly in defined ecological contexts, including newborn cetaceans that forgo sleep, the Arctic tern that suppresses sleep during the mating season, and frigate birds that have reduced and unilateral sleep during prolonged flight (*Lesku et al., 2012*; *Lyamin et al., 2005*; *Rattenborg et al., 2016*). It will be

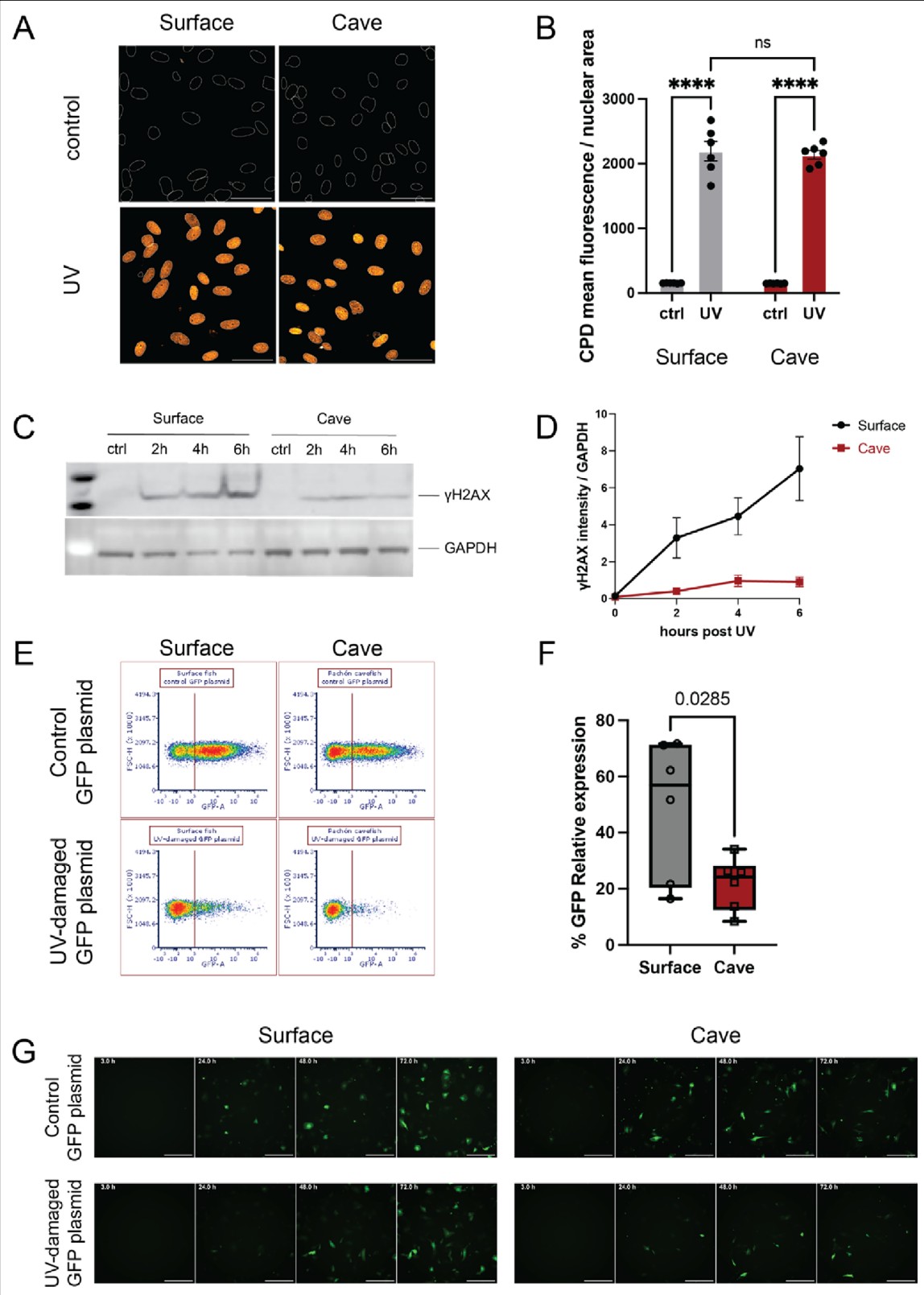

**Figure 4.** Pachón cavefish-derived cells exhibit a lower UV-induced DNA damage response and repair compared to surface fish. (**A, B**) Immunostaining of cyclobutene pyrimidine dimer (CPD) shows a similar DNA damage level induced by UV in surface fish and Pachón cavefish embryonic fibroblasts. White circles indicate the nuclear area by DAPI staining. Orange indicates CPD. Scale bar, 40 μm. *P*-values were determined by two-way ANOVA: $F=0.09703$, $p=0.7586$. ns $p=0.6404$, ****$p<0.0001$. (**C, D**) Western blot of γH2AX indicates a diminished DNA damage response in Pachón cavefish

*Figure 4 continued on next page*

*Figure 4 continued*

embryonic fibroblasts compared to surface fish cells. (**E, F**) Flow cytometry images and quantification for host cell reactivation assays in surface fish and Pachón cavefish embryonic fibroblasts. Red line sets the green fluorescent protein (GFP) positive signal threshold. *P*-values were determined by unpaired *t*-test. (**G**) Representative GFP images for host cell reactivation assays in surface fish and Pachón cavefish embryonic fibroblasts. Scale bars, 500 μm.

The online version of this article includes the following source data and figure supplement(s) for figure 4:

**Source data 1.** Original gels for data presented in *Figure 4*.

**Source data 2.** Original western blots for data presented in *Figure 4*.

**Figure supplement 1.** Pachón cavefish-derived cells exhibit a lower UV-induced DNA damage response and repair compared to surface fish.

**Figure supplement 1—source data 1.** Original gels for data presented in *Figure 4—figure supplement 1*.

**Figure supplement 1—source data 2.** Original western blots for data presented in *Figure 4—figure supplement 1*.

highly informative to investigate the presence of DNA damage and other markers of sleep loss within a natural ecological context and in evolutionary models with altered sleep.

Multiple studies have now revealed changes in sleep, locomotor behavior, and DNA damage across different species of cavefish (*Keene et al., 2024*). The adaptive value of these changes in a natural setting, and their impact on longevity, remain poorly understood. Additional experiments are necessary to address the cellular relationship between sleep and DNA damage. In zebrafish, sleep loss induces DNA damage, while inducing DNA damage promotes sleep (*Zada et al., 2021*). These findings suggest that DNA damage contributes to sleep homeostasis. We have previously shown that cavefish rebound following sleep deprivation, suggesting they maintain an intact sleep homeostat (*McGaugh et al., 2020b*).

Our cellular analysis further revealed a muted DDR in cavefish, which likely contributes to the increased DNA damage noted in vivo. Since photolyases are primarily utilized for the repair of UV-induced cyclobutene pyrimidine dimers, and their repair processes are also dependent on light input (*Liu et al., 2011*), it is plausible that these genes were not favored by natural selection in cavefish, leading to accumulated mutations and a loss of functional DNA damage response. This hypothesis is supported by findings that Somalian cavefish have lost critical DNA repair enhancers needed for an induced DDR (*Zhao et al., 2018*). The fact that Mexican cavefish retain some capacity for light-induced DNA repair might be attributed to their relatively recent divergence from their surface-dwelling counterparts, estimated to be less than 200,000 years ago (*Fumey et al., 2018*; *Herman et al., 2018*). Although our findings are consistent with studies on DNA repair in other cavefish species, they contrast with previous research suggesting an increased DNA repair function in certain *Astyanax* cavefish populations (*Beale et al., 2013*). While increased expression of some DNA repair genes is also observed in our study (*Figure 4—figure supplement 1*), our cellular assays demonstrate that this does not equate to enhanced DNA repair activity. However, our analysis focused only on two different cell lines (embryonic fibroblasts and liver-derived cells), while the previous study looked at DNA repair in fins (*Beale et al., 2013*). Further research will be required to resolve these differences and fully understand DNA repair dynamics in cavefish.

In *Drosophila* and mice, both acute and chronic sleep loss is associated with reduced longevity, and it is postulated that ROS-associated gut dysregulation leads to death in chronically sleep-deprived individuals (*Vaccaro et al., 2020*). Therefore, it is interesting that cavefish do not exhibit clear signs of accelerated aging compared to surface fish (*Medley et al., 2022*; *Riddle et al., 2018a*). The lifespan of surface and cave populations of *A. mexicanus* is reported to exceed 20 years, largely preventing the use of longevity as a readout for aging (Rohner, personal communication). However, DNA damage and aging-associated transcriptional changes can provide a proxy for biological aging. The findings that the transcriptional architecture of cavefish does not vary to the degree of surface fish indicate a form of genomic stability or a decelerated aging process, which could be a focal point for future research into the mechanics of aging and its relationship with sleep and DNA repair. In addition to sleep loss, cavefish have fatty livers, reductions in heart regeneration, and chronically elevated blood glucose (*Aspiras et al., 2015*; *Riddle et al., 2018a*). These findings raise the possibility that cavefish may have evolved a broad range of resilience mechanisms to biological stress. Consistent with this notion, cavefish have evolved a reduced metabolic rate and elevated metabolites associated with hypoxia and longevity (*Medley et al., 2022*). Many of these features, such as reduced metabolic

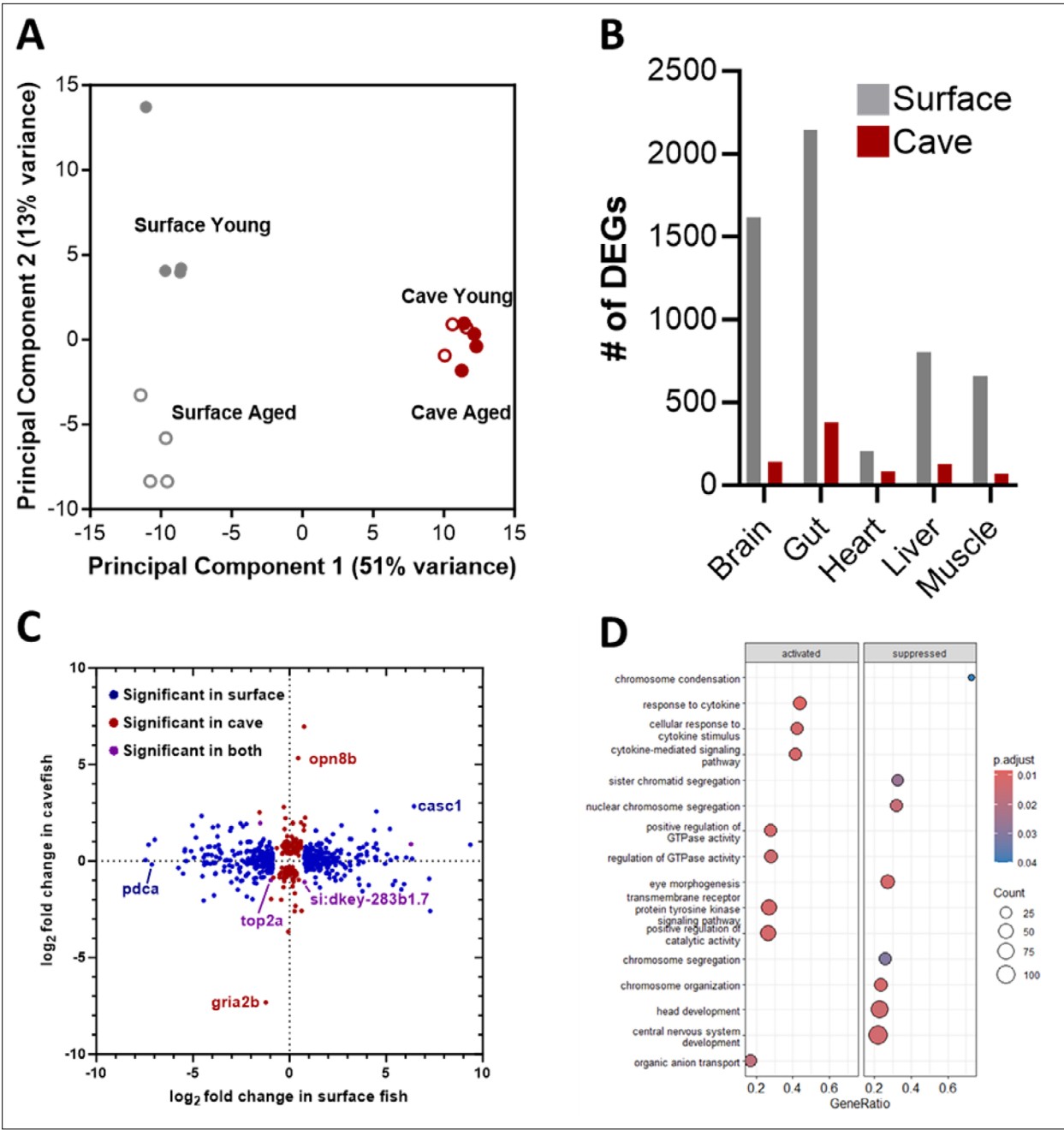

**Figure 5.** Transcriptional response to aging is diminished in cavefish across tissues. (**A**) Multidimensional scaling plot plotting the distances in principal component space between the brain samples. PC1 two-way ANOVA: (Treatment) $F_{1,11}$=4.209, $p$=0.0648, (Population) $F_{1,11}$=2133, $p$<0.0001, (Interaction) $F_{1,11}$=0.029, $p$=0.867 PC2 two-way ANOVA: (Treatment) $F_{1,11}$=16.83, $p$=0.0018, (Population) $F_{1,11}$=0.0002, $p$<0.99, (Interaction) $F_{1,11}$=19.37, $p$=0.0011 (**B**) Number of differentially expressed genes in the aged condition across tissues. (**C**) Bi-directional volcano plot depicting differences in gene expression between young and aged brains. (**D**) Dot plot visualizing the top 8 activated and suppressed gene ontology terms in aged cavefish brains resulting from gene set enrichment (GSE) analysis.

The online version of this article includes the following figure supplement(s) for figure 5:

**Figure supplement 1.** Aging-induced changes in gene expression.

**Figure supplement 2.** Gene set enrichment (GSEA) analyses of aging surface fish and cavefish tissues.

rate, are present in other long-lived organisms, including the naked mole-rat (*Heterocephalus glaber*) and the cave olm (*Proteus anguinus*) (*Lunghi and Bilandžija, 2022*). Comparing differences between surface fish and cavefish, or similarities between cavefish and other long-lived models, may provide a system to study resilience to biological stress.

Taken together, these findings suggest that cavefish can be used as a model to study the evolved loss of DDR and biological resilience. Growing genetic and genomic tools in cavefish, including multiple chromosome-level sequenced genomes and single-cell RNA sequencing atlases, may allow for the identification of markers of selection for DNA repair and cell-type-specific transcriptional changes in sleep-regulating neurons. In addition, the study of DNA damage and ROS in additional populations of cavefish provides the opportunity to identify multiple mechanisms of DNA damage response.

## Materials and methods

### Fish husbandry

Fish used in behavioral, immunohistochemistry, and UV-B experiments were generated and raised at Texas A&M University in dedicated aquarium facilities, with water temperature maintained at 23 °C on a 14 hr light: 10 hr dark schedule as previously described (*Kozol et al., 2023b*). To stimulate breeding, water temperature was raised by ~2 °C with a submersible heater, and fish were fed frozen bloodworms to satiation 2–3 times daily (*Elipot et al., 2014*). Following breeding, embryos were collected from tanks and raised in a temperature-controlled incubator under the same temperature and light cycle until used for experiments. Fish used for tissue dissections were collected from aquarium facilities at Florida Atlantic University, maintained under the same conditions. Fish and embryos used for cell line derivation were collected from the cavefish facility at the Stowers Institute for Medical Research, maintained under similar conditions, except for the food. Adult fish were fed with Mysis shrimp once in the morning and Gemma 800 (Skretting, Gemma Silk) once in the afternoon. Fish in the breeding cycle were fed with Gemma 800 once at noon.

### Fibroblast cell line derivation

Embryos were collected after spawning and were dechorionated before 14 hr post fertilization (hpf). Embryos were then washed in washing media (PBS supplemented with 50 U/ml Penicillin +0.05 mg/ml Streptomycin) for 30 min three times. Five embryos were transferred to a sterile 1.5 ml tube with 0.5 ml of wash media, where the yolk sac of the embryos was removed by pipetting up and down ~10 times with a 200 µl pipette tip. Tissues in 1.5 ml tubes were then centrifuged at 1200 g for 2 min at room temperature (RT) and the supernatant was discarded carefully. Next, 300 µl of TrypLE (Thermo Fisher Scientific, #12604021) was added to each 1.5 ml tube and incubated at 28 °C for 5 min in a thermomixer at 800 g. Once the embryos were dissociated, embryo tissues were pipetted up and down several times, followed by centrifugation at 1200 g for 4 min. Cell pellets were resuspended in 400 µl growth media containing 15% fetal bovine serum (FBS) (Cytiva, #SH30071.02E), 0.8 mM calcium chloride, 1 µg/ml human insulin solution (Sigma, #I9278), and 1 X L-Glutamine (Thermo Fisher Scientific, #25030081) in L-15 (Corning, #10–045-CV), and transferred to 1 well of a 48 well plate coated with 0.1% gelatin in advance. Cells were maintained and incubated with growth media in the incubator at 28 °C without $CO_2$ and were observed every day with media changes 3 times per week (400 µl per well). Cells usually reached confluency 7–10 days after derivation. Cells were passaged once a week at a 1:4 to 1:10 ratio and expanded for cryopreservation in freezing media (90% FBS +10% DMSO) at $2\times10^6$ per vial.

### Host cell reactivation (HCR) assay

Experiments were designed and protocols were modified based on two previous studies (*Domankevich et al., 2018*; *Tian et al., 2019*). An amount of 25 µl of pmaxGFP plasmid from Lonza Amaxa cell Nucleofector Kit V (Lonza, #VCA-1003) at a concentration of ~0.2 µg/µl was aliquoted into a 10 cm plate and irradiated with 600 $J/m^2$ of UV-C light with the lid open. The same batch of UV-treated plasmids was used for testing all species to avoid batch-to-batch variation. Next, $1.2\times10^6$ cells were co-transfected with 6 µg of treated or untreated pmaxGFP. For each reaction, 82 µl base media and 18 µl of supplement was used to make the Nucleofector Solution. Cell pellets were resuspended carefully in 100 µl RT Nucleofector Solution per sample with 5 µg treated or untreated pmaxGFP added,

then transferred into a certified cuvette. It was necessary for the samples to cover the bottom of the cuvette without air bubbles. Transfections were carried out with program T-027 using Amaxa Nucleofector II. Cuvettes were taken out of the holder once the program was finished. An amount of 20 µl of the mixture was added to 24-well plates with 1 ml of cell media and imaged every 1.5 hr for 3 days using a Nikon ECLIPSE Ti2 microscope with a 10 X lens; the rest of the cells were added to 6-well plates with 3 ml of cell media for flow cytometry analysis.

## Flow cytometry

Cells were harvested 50 hr post-transfection. The cell pellet collected from each well of a 6-well plate was resuspended in 500 µl 1 X PBS. An amount of 1 µl of Ghost Dye solution (Cytek Bioscience, #13–0865 T100) was added to each 500 µl cell suspension, vortexed immediately, and was then incubated for 30 min at 2–8°C protected from light. After incubation, the cell suspension was spun at 400 g for 5 min at RT and the supernatant was discarded. Next, 500 µl of media was added to the cell pellets to prepare for flow cytometry using Cytek Aurora. The %GFP relative expression (%RE) was calculated using $F = N \times MFI/S$, where N is the total number of live cells appearing in the positive region for GFP, MFI is the mean fluorescence intensity of the N cells, and S is the total number of live cells; %RE = $F_t/F_u$, where $F_t$ is the F of cells transfected with treated plasmids and $F_u$ is the F of cells transfected with undamaged plasmids. Live cells were identified based on altered forward and side scatters.

## Immunohistochemistry

For immunostaining in fish, larvae aged 6 dpf were fixed at ZT1 in a 1 X Phosphate Buffer Solution (PBS) with 4% Paraformaldehyde and 0.1% Tween 20 for 6 hr on ice as previously described (*Jaggard et al., 2020*; *Kozol et al., 2023a*). After fixation, samples were rinsed three times in PBS with 0.1% Tween 20 (PBT) at RT, with 10 min between rinses. Samples were incubated overnight at 4 °C in 1:500 anti-γH2AX (Genetex, #GTX127342) with 0.1% PBT and 2% Bovine Serum Albumin (BSA). After overnight incubation, samples were rinsed 3 times in 0.1% PBT at RT, with 10 min between rinses. Next, samples were incubated in 1:500 Goat anti-rabbit 488 (Abcam, #ab150077) and 1 µg/ml DAPI (Thermo Fisher Scientific, #D1306) in 0.1% PBT. Finally, samples were rinsed again 3 times in 0.1% PBT at RT with 10 min between rinses, stored overnight at 4 °C, and then imaged on a Nikon A1 confocal microscope with a 20 X water immersion lens.

For vimentin staining in cell culture, cells were fixed with 4% paraformaldehyde in PBS for 15 min at RT. After fixation, cells were rinsed 3 times in PBS at RT, with 5 min between rinses. After fixation, cells can be stored for up to 1 week at 4 °C. Fixed cells were permeabilized with 0.1% Triton in PBS for 30 min at RT, then blocked with 20% FBS diluted in 0.1% PBT for 1 hr at RT. Samples were then incubated overnight at 4 °C with 1:200 anti-vimentin antibody (Sigma, #V5255) in blocking buffer. After overnight incubation, samples were rinsed 3 times in 0.1% PBT at RT with 5 min between rinses. Next, samples were incubated for 1 hr at room temperature in darkness with 1:300 Donkey Anti-Mouse IgG (H+L) 568 (Biotium, #20105) and 1 µg/ml DAPI (Sigma, #10236276001) in 5% FBS and 0.1% PBT. Finally, samples were rinsed again three times in 0.1% PBT with 5 min between rinses, rinsed once with PBS, stored overnight at 4 °C, and then imaged on a Nikon ECLIPSE Ti2 confocal microscope with a 40 X water immersion lens.

For CPD staining in cell culture, cells were plated at $2 \times 10^5$ per well in a glass-bottom dish (Cellvis, #D35C4-20-1.5-N) 2 days before the experiment. Cells were washed once with PBS, and cell media was removed before exposure to 100 J/m² UV-C (Stratalinker UV Crosslinker, Model 2400, #400075). Cells were fixed 5 min after UV-C treatment using 4% paraformaldehyde in PBS for 15 min at RT. After fixation, cells were rinsed three times in PBS at RT with 5 min between rinses. After fixation, cells can be stored up to 1 week at 4 °C. Fixed cells were permeabilized with 0.1% Triton in PBS for 30 min at RT, followed by two rinses in PBS with 5 min between rinses. Next, 2 M Hydrochloric acid (HCl) was used to denature cellular DNA for 30 min at RT, followed by five rinses in 0.1% PBT with 5 min between rinses. After this, cells were blocked with 20% FBS diluted in 0.1% PBT for 1 hr at RT. Samples were then incubated overnight at 4 °C with 1:1000 anti-CPD antibody (TDM-2, Cosmo Bio, #CAC-NM-DND-001) in blocking buffer. After overnight incubation, samples were rinsed three times in 0.1% PBT at RT with 5 min between rinses. Next, samples were incubated for 1 hr in darkness with 1:300 Donkey Anti-Mouse IgG (H+L) 568 (Biotium, #20105) and 1 µg/ml DAPI (Sigma, #10236276001) in 5%

FBS and 0.1% PBT. Finally, samples were rinsed again three times in 0.1% PBT with 5 min between rinses, rinsed once with PBS, stored overnight at 4 °C, and then imaged on a Nikon ECLIPSE Ti2 confocal microscope with a 40 X water immersion lens.

## Western blot analysis

Fish fibroblast cells and liver-derived cells (*Krishnan et al., 2022*) were plated at $5\times10^4$ per well in 6-well plates 4 days before the experiment. Cell media was removed before cells were exposed to 3000 J/$m^2$ UV-C. Fresh cell growth media was added to the cells and incubated for 2, 4, or 6 hr. Lysis buffer was made with 1 X protease inhibitor (25 X stock, Sigma, #11873580001) and 1 X phosphatase inhibitor (100 X stock, Cell Signaling, #5870 S) in RIPA buffer (Thermo Fisher Scientific, #89900). Next, 100 μl lysis buffer was added to each well, and cells were detached by cell scraper (VWR INTERNATIONAL, #10062–904). Samples were fully lysed by pipetting up and down ~5 times with a fine needle syringe (BD Medical, #328438) and vortexed for 10 s. The supernatant was collected as protein fractions after two rounds of centrifugation at 12,000 g for 20 min at 4 °C. Protein concentrations were determined by bicinchoninic acid (BCA) protein assay (Thermo Fisher Scientific, #23227).

Protein samples were loaded onto the protein gel (Thermo Fisher Scientific, #NP0323BOX) at ~10–13 μg per lane. Protein gels were run in MOPS buffer (Thermo Fisher Scientific, #NP0001) at 130 V for 80 min, then transferred to PVDF membrane (Sigma, #IPVH00010) at 4 °C at 215 mA for 50 min. PVDF membranes were blocked in blocking buffer (LI-COR, #927–80001) for an hour at RT, then incubated with primary antibody overnight at 4 °C in 1:1000 anti-γH2AX (Genetex, #GTX127342) and 1:1000 anti-GAPDH (Proteintech, #60004–1-Ig) with blocking buffer. On the next day, PVDF membranes were washed three times in 0.1% TBST with 5 min between rinses and incubated with 1:10,000 Donkey anti-mouse (LI-COR, #926–32212) and anti-rabbit (LI-COR, #926–68073) secondary antibody at RT for an hour. Finally, samples were rinsed again three times in 0.1% TBST with 10 min between rinses, and then imaged and quantified on LI-COR Odyssey CLx with Image Studio v5.2 software.

## ROS imaging

ROS imaging was implemented based on previous protocols used in *Drosophila* (*Vaccaro et al., 2020*). Larvae aged 6 dpf were euthanized at ZT1 by immersion in ice-chilled aquarium water for 30 min. Following euthanasia, samples were incubated in 60 μM dihydroethidium (DHE; Thermo Fisher Scientific, D11347) for 5 min at RT, covered in foil to protect them from light. Next, samples were rinsed for 5 min in 1 X PBS, then mounted in 2% low meltingpoint agarose (Sigma-Aldrich, A9414), and the entire gut was imaged on a Nikon A1R confocal microscope with a 10 X air objective, with a slice thickness of 4 μm. After interaction with superoxide radicals, DHE is converted into 2-hydroxy ethidium, which is optimally excited at 480 nm and fluoresces red (wavelength >550 nm); therefore, the sample was excited with a 488 nm laser and imaged with a detector in the 570–616 nm range (*Kumar and Gullapalli, 2024*). ROS in the gut was quantified by calculating a z-projection (calc: Average Intensity) in ImageJ (NIH, v1.54f) and manually drawing ROIs around anatomically defined regions of the gut (*Riddle et al., 2018b*).

## Sleep experiments

Sleep experiments used the previously described methodology (*Jaggard et al., 2019*). For induction of DNA damage, larvae aged 5 dpf were transferred to individual wells of a 24-well plate at ZT7–8 (VWR, 82050–892), and acclimated overnight to the testing environment. At ZT0 the next day, larvae were subjected to either 60 s of control white light or 30 or 60 s of UV-B light in a UV-Crosslinker cabinet (Spectro-UV, XL-1500) fitted with UV bulbs with a spectral peak at 306 nm and a power output of 32.3 μW/$cm^2$ at 1 meter. (UV-B; Spectro-UV, BLE-1T158). Following treatment, well plates were immediately returned to the testing environment and filmed from above using a USB camera (Basler, acA1300-200) fitted with a 16 mm fixed focal length lens (Edmund Optics, 67–714) and an infrared pass filter (Edmund Optics, 65–796) to ensure consistent image quality after the day/night transition. Larvae were lit from below using infrared light strips (850 nm), diffused through a custom-made white acrylic light box (TAP Plastics).

Locomotor behavior was tracked in Ethovision XT (Noldus), and frame-by-frame velocity data was exported and analyzed using Python, with sleep defined as 60 s or more of consolidated immobility

(*Jaggard et al., 2020*). A velocity cut-off of 6 mm/s was used to distinguish active swimming from passive drift.

## RNA extraction and sequencing

For RNA sequencing on fish fibroblast cells, cells were detached with TrypLE and washed once with cell media. Cell pellets were collected after centrifugation at 1000 g for 5 min, snap frozen in liquid nitrogen, and stored at –80 °C prior to RNA extraction. For UV RNA sequencing experiments, larvae were treated identically to the sleep experiments until 1.5 hr after UV exposure, at which point they were transferred to 1.5 ml centrifuge tubes, with four larvae pooled into each tube. The larvae were euthanized by chilling on ice before immediately proceeding to RNA extraction.

For RNA sequencing experiments on adults, dissections were performed at Florida Atlantic University. Briefly, tissue was dissected between ZT0 and ZT3 from at least four fish at each age and population group. Tissue was immediately flash-frozen in liquid nitrogen. To extract RNA, 1 ml of TRIzol was added to each sample. Samples were homogenized, and then 200 μl chloroform was added to each tube, followed by vigorous shaking. Samples were incubated on ice for 15 min, then phase separated by centrifuging at 12,000 g for 15 min at 4 °C. The resulting aqueous phase of the liquid was transferred to a fresh 1.5 ml tube, and then RNA was precipitated out by mixing with 0.5 ml isopropanol. Samples were incubated on ice for 10 min, then centrifuged at 12,000 g for 10 min at 4 °C. The supernatant was removed by pouring and lightly shaking the tube, then the resulting pellet was washed by the addition of 1 ml 70% EtOH while vigorously flicking the tube. The samples were then centrifuged at 7500 g for 10 min at 4 °C. The supernatant was removed by pouring and lightly shaking the tube, and the samples were air-dried upside down for 10 min. Finally, the pellet was redissolved in 100 ul of RNAse-free $H_2O$.

## RNA sequencing data analysis

RNA sequencing for the UV and Aging experiments were performed by Novogene, on an Illumina NovaSeq X Plus. Sample quality control, library preparation, and sequencing were performed by Novogene. Raw reads received from Novogene were mapped against the *Astyanax mexicanus* reference genome (version 2.0, GenBank Accession Number: GCA_000372685.2) using the splice-aware mapper STAR (*Dobin et al., 2013*) to generate raw counts. Annotations were extracted from the *A. mexicanus* annotation file from Ensembl (Astyanax_mexicanus-2.0.108.gtf). Subsequent analysis was performed in RStudio (v4.3.0) using the differential expression testing software DESeq2 (v1.40.2) (*Love et al., 2014*). RNA sequencing for the fibroblast cell lines was performed on an Illumina NextSeq 500.

Data from the UV-B experiment and the aging experiment were analyzed using the same analysis pipeline, with each tissue type from the aging experiment analyzed separately. First, a DESeq object was created from the raw counts matrix and processed using the DESeq() command, which estimates size factors and dispersion, and fits the data to a negative binomial GLM. Normalized count values used in downstream analysis were generated using the counts() function (normalized = T). Sample variability for each group was visualized by first performing a variance stabilizing transformation using the vst() function, then generating a PCA plot from the resulting object with the plotPCA() function. To identify genes which were differentially expressed in response to UV-B treatment (or aging), each population treatment subset was reanalyzed as above, so that the effects of treatment on each population were considered separately. An adjusted *p*-value <0.05 was used to determine a significant response to treatment.

## Gene Ontology (GO) analysis

Gene ontology pathway analyses were performed using the clusterProfiler package (v4.8.3) in R. For gene set enrichment analyses, genes were first ranked according to the magnitude of their response; the ranking value was calculated as $-\log_{10}(pval)/sign(\log_2 FC)$, to account for both direction and magnitude of response. The resulting ranked list was processed using the gseGO() function (options: ont='BP", keyType='SYMBOL', pvalueCutoff = 0.05, OrgDB = org.Dr.eg.db, pAdjustMethod='BH'). For visualization purposes, similar GO terms were grouped together using the simplify() function (options: by='p.adjust', select_fun = min), and visualized using the dotplot() function. For the UV-B experiment, since there were many overlapping GO terms, the top 10 unique terms in each direction were plotted.

## Quantitative PCR

Adult fins were collected at Zeitgeber or Circadian time 8. 5 dpf larvae were collected at Zeitgeber or Circadian time 2, 6, 10, and 14. Samples were then homogenized and total RNA extracted as above. cDNA was synthesized from 500 ng RNA using iScript cDNA synthesis kit (BIO-RAD, #1708891). Approximately 50 ng of cDNA was used for quantitative PCR (qPCR) using Perfecta SYBR Green with Low ROX (Quantabio, #95074–012) with a QuantStudio 5 Real-Time PCR System. Specificity of each amplicon and cDNA final concentration was optimized via analysis of post-reaction dissociation curves, validating a single amplicon for each set of primers. Analysis was conducted using the ΔΔCt method. All samples were run in 3–4 replicates and normalized to the housekeeping gene *rpl13a*. Primer sequences used are as follows:

*cpdp* (ENSAMXG00000001885): FW: 5'- GGCCTCTCCTAAGCTGGAGT –3'
RV: 5'- GTCCACAGGTGGGAATTCAG -3' *ddb2* (ENSAMXG00000000525):
FW: 5'- AAGCTGCACAAAGCCAAAGT-3'
RV: 5'- AGACGATGTTGCCACTAGCC -3'.
*rpl13a* (ENSAMXG00000033532):
FW 5'- CGCAACAAATTGAAGTACCTG -3'
RV: 5'- GGTTCGTGTTCATCCTCTTG -3'

## Acknowledgements

We would like to thank the help and support provided by facility cores at Stowers Institute; specifically, the cavefish team for fish husbandry, KyeongMin Bae and Jose Emmanuel Javier for flow cytometry, and Di Wu for data analysis. This work was supported by the National Institute of Health R24OB030214 to WW, NR, and ACK and R21 NS122166 to ACK and US-Israel Binational Science Foundation Award 2021177 to ACK and LA.

## Additional information

### Funding

| Funder | Grant reference number | Author |
|--------|------------------------|--------|
| National Institutes of Health | R24OB030214 | Alex C Keene |
| National Institutes of Health | NS122166 | Alex C Keene |
| Israel Science Foundation | 2021177 | Lior Appelbaum |

The funders had no role in study design, data collection and interpretation, or the decision to submit the work for publication.

### Author contributions

Evan Lloyd, Conceptualization, Formal analysis, Methodology, Writing – original draft, Writing – review and editing; Fanning Xia, Formal analysis, Visualization, Writing – original draft, Writing – review and editing; Kinsley Moore, Formal analysis, Writing – review and editing; Carolina Zertuche Mery, Investigation, Methodology, Writing – original draft, Writing – review and editing; Aakriti Rastogi, Conceptualization, Resources, Funding acquisition, Investigation, Methodology, Writing – original draft, Writing – review and editing; Robert A Kozol, Conceptualization, Funding acquisition, Investigation, Visualization, Methodology, Writing – original draft, Writing – review and editing; Olga Kenzor, Conceptualization, Resources, Data curation, Investigation, Methodology, Writing – original draft, Project administration, Writing – review and editing; Wesley Warren, Conceptualization, Data curation, Investigation, Methodology, Writing – original draft, Writing – review and editing; Lior Appelbaum, Conceptualization, Resources, Data curation, Formal analysis, Visualization, Writing – original draft, Writing – review and editing; Rachel L Moran, Conceptualization, Investigation, Writing – original draft, Writing – review and editing; Chongbei Zhao, Conceptualization, Funding acquisition, Investigation, Methodology, Writing – original draft, Writing – review and editing; Erik

R Duboue, Conceptualization, Resources, Supervision, Funding acquisition, Investigation, Methodology, Writing – original draft, Writing – review and editing; Nicolas Rohner, Conceptualization, Supervision, Funding acquisition, Methodology, Writing – original draft, Project administration, Writing – review and editing; Alex C Keene, Resources, Formal analysis, Supervision, Funding acquisition, Visualization, Methodology, Writing – original draft, Project administration, Writing – review and editing

### Author ORCIDs
Evan Lloyd ⓘ https://orcid.org/0000-0002-8473-3636
Carolina Zertuche Mery ⓘ https://orcid.org/0000-0002-9245-0812
Chongbei Zhao ⓘ https://orcid.org/0000-0002-7224-5600
Erik R Duboue ⓘ https://orcid.org/0000-0003-3303-5149
Nicolas Rohner ⓘ https://orcid.org/0000-0003-3248-2772
Alex C Keene ⓘ https://orcid.org/0000-0001-6118-5537

### Ethics
This study was performed in strict accordance with the recommendations in the Guide for the Care and Use of Laboratory Animals of the National Institutes of Health. All of the animals were handled according to approved institutional animal care and use committee (IACUC) protocols 2024-0277 at Texas A&M. The protocol was approved by the Committee on the Ethics of Animal Experiments of Texas A&M.

Reviewer #1 (Public review): https://doi.org/10.7554/eLife.99191.3.sa1
Reviewer #2 (Public review): https://doi.org/10.7554/eLife.99191.3.sa2
Reviewer #3 (Public review): https://doi.org/10.7554/eLife.99191.3.sa3
Author response https://doi.org/10.7554/eLife.99191.3.sa4

## Additional files

### Supplementary files
MDAR checklist

Supplementary file 1. UV-B Treatment gene expression raw data.

Supplementary file 2. Aging gene expression raw data.

### Data availability
Raw data from RNASeq experiments have been deposited in GEO as a SuperSeries, under accession code GSE272697. Data from UV-B experiments has been deposited under accession code GSE271729. Data from aging experiments has been deposited under accession code GSE272611. Raw count data, and results of differential gene expression analysis, have been uploaded as *Supplementary file 1* (UV) and *Supplementary file 2* (aging).

The following datasets were generated:

| Author(s) | Year | Dataset title | Dataset URL | Database and Identifier |
|---|---|---|---|---|
| Keene AC | 2024 | Cavefish | https://www.ncbi.nlm.nih.gov/bioproject/?term=GSE272697 | NCBI BioProject, GSE272697 |
| Keene AC | 2024 | Effects of UV-B radiation on whole-body gene expression of surface and cave Astyanax mexicanus (Mexican tetra) | https://www.ncbi.nlm.nih.gov/bioproject/?term=%20GSE271729 | NCBI BioProject, GSE271729 |
| Keene AC | 2024 | Effects of aging on gene expression in multiple tissues in surface and cave Astyanax mexicanus (Mexican tetra) | https://www.ncbi.nlm.nih.gov/bioproject/?term=%20GSE272611 | NCBI BioProject, GSE272611 |

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
