## [Editor Report · eLife Assessment]

Lloyd et al. used an evolutionary comparative approach to study DNA damage repair associated with low sleep duration in Astyanax mexicanus, highlighting how the cavefish population has evolved a reduced DNA damage response. The results presented here have **important** implications. Their results are generally **solid**, however the evidence suggesting that sleep differences are linked to DNA damage response is missing and this hypothesis remains to be fully tested.

---

## [Referee Report · Reviewer #1 (Public review)]

Summary:

Lloyd et al employ an evolutionary comparative approach to study how sleep deprivation affects DNA damage repair in Astyanax mexicanus, using the cave vs surface species evolution as a playground. The work shows, convincingly, that the cavefish population has evolved an impaired DNA damage response both following sleep deprivation or a classical paradigm of DNA damage (UV).

Strengths:

The study employs a thorough multidisciplinary approach. The experiments are well conducted and generally well presented.

Weaknesses:

Having a second experimental mean to induce DNA damage would strengthen and generalise the findings.

Overall, the study represents a very important addition to the field. The model employed underlines once more the importance of using an evolutionary approach to study sleep and provides context and caveats to statements that perhaps were taken a bit too much for granted before. At the same time, the paper manages to have an extremely constructive approach, presenting the platform as a clear useful tool to explore the molecular aspects behind sleep and cellular damage in general. The discussion is fair, highlighting the strengths and weaknesses of the work and its implications.

Comments on revisions:

I was pretty happy with the previous version of the manuscript already and the authors have made all the minor corrections I had suggested so I don't have much to add. The main "weakness", if at all, is that the story would benefit from a secondary stressor (other than UV) but I understand the authors see this more as a long term development than just an addition to this particular paper, which is fair enough.

I don't have any further recommendations. I think this model system is really important for the sleep field and offers a completely new and important perspective to its evolution and function.

---

## [Referee Report · Reviewer #2 (Public review)]

The manuscript investigates the relationship between sleep, DNA damage, and aging in the Mexican cavefish (Astyanax mexicanus), a species that exhibits significant differences in sleep patterns between surface-dwelling and cave-dwelling populations. The authors aim to understand whether these evolved sleep differences influence the DNA damage response (DDR) and oxidative stress levels in the brain and gut of the fish.

Summary of the Study:

The primary objective of the study is to determine if the reduced sleep observed in cave-dwelling populations is associated with increased DNA damage and altered DDR. The authors compared levels of DNA damage markers and oxidative stress in the brains and guts of surface and cavefish. They also analyzed the transcriptional response to UV-induced DNA damage and evaluated the DDR in embryonic fibroblast cell lines derived from both populations.

Strengths of the Study:

Comparative Approach: The study leverages the unique evolutionary divergence between surface and cave populations of A. mexicanus to explore fundamental biological questions about sleep and DNA repair.

Multifaceted Methodology: The authors employ a variety of methods, including immunohistochemistry, RNA sequencing, and in vitro cell line experiments, providing a comprehensive examination of DDR and oxidative stress.

Interesting Findings: The study presents intriguing results showing elevated DNA damage markers in cavefish brains and increased oxidative stress in cavefish guts, alongside a reduced transcriptional response to UV-induced DNA damage.

Weaknesses of the Study:

Link to Sleep Physiology: The evidence connecting the observed differences in DNA damage and DDR directly to sleep physiology is not convincingly established. While the study shows distinct DDR patterns, it does not robustly demonstrate that these are a direct result of sleep differences.

Causal Directionality: The study fails to establish a clear causal relationship between sleep and DNA damage. It is possible that both sleep patterns and DDR responses are downstream effects of a common cause or independent adaptations to the cave environment.

Environmental Considerations: The lab conditions may not fully replicate the natural environments of the cavefish, potentially influencing the results. The impact of these conditions on the study's findings needs further consideration.

Photoreactivity in Albino Fish: The use of UV-induced DNA damage as a primary stressor may not be entirely appropriate for albino, blind cavefish. Alternative sources of genotoxic stress should be explored to validate the findings.

Assessment of the Study's Achievements:

The authors partially achieve their aims by demonstrating differences in DNA damage and DDR between surface and cavefish. However, the results do not conclusively support the claim that these differences are driven by or directly related to the evolved sleep patterns in cavefish. The study's primary claims are only partially supported by the data.

Impact and Utility:

The findings contribute valuable insights into the relationship between sleep and DNA repair mechanisms, highlighting potential areas of resilience to DNA damage in cavefish. While the direct link to sleep physiology remains unsubstantiated, the study's data and methods will be useful to researchers investigating evolutionary biology, stress resilience, and the molecular basis of sleep.

Comments on revisions:

The manuscript should tone down claims of a direct causal relationship between sleep differences and DDR outcomes, acknowledging the possibility that both are independent or downstream adaptations to the cave environment. To strengthen the study, the authors should adopt additional genotoxic stressors, such as chemical agents (e.g., cisplatin or hydrogen peroxide) or physical stress (e.g., ionizing radiation), to validate findings beyond UV-induced DNA damage, which may not be ideal for albino cavefish. Explicitly discussing the influence of laboratory conditions, such as water quality, lighting, and diet, on oxidative stress and DDR phenotypes, and comparing lab-reared and wild-caught fish if feasible, would bolster ecological relevance. The study should clarify that the current data do not establish a causal link between sleep and DNA damage, instead proposing this as a hypothesis for future research. Expanding the evolutionary context by linking DDR differences to other cavefish traits, such as metabolic efficiency or hypoxia tolerance, could provide a more integrative perspective. Additionally, proposing future experiments involving pharmacological or behavioral manipulation of sleep, as well as incorporating comparative genomics or transcriptomics to identify DDR-related genetic adaptations, would enhance the study's depth.

---

## [Referee Report · Reviewer #3 (Public review)]

Lloyd, Xia et al. utilised the existence of surface-dwelling and cave-dwelling morphs of Astyanax mexicanus to explore a proposed link between DNA damage, aging, and the evolution of sleep. Key to this exploration is the behavioural and physiological differences between cavefish and surface fish, with cavefish having been previously shown to have low levels of sleep behaviour, along with metabolic alterations (for example chronically elevated blood glucose levels) in comparison to fish from surface populations. Sleep deprivation, metabolic dysfunction and DNA damage are thought to be linked, and to all contribute to aging processes. Given that cavefish seem to show no apparent health consequences of low sleep levels, the authors suggest that they have evolved resilience to sleep loss. Furthermore, as extended wake and loss of sleep is associated with increased rates of damage to DNA (mainly double-strand breaks) and sleep is linked to repair of damaged DNA, the authors propose that changes in DNA damage and repair might underlie the reduced need for sleep in the cavefish morphs relative to their surface-dwelling conspecifics.

To fulfil their aim of exploring links between DNA damage, aging, and the evolution of sleep, the authors employ methods that are largely appropriate, and comparison of cavefish and surface fish morphs from the same species certainly provides a lens by which cellular, physiological and behavioural adaptations can be interrogated. Fluorescence and immunofluorescence are used to measure gut reactive oxygen species and markers of DNA damage and repair processes in the different fish morphs, and measurements of gene expression and protein levels are appropriately used. However, although the sleep tracking and quantification employed is quite well established, issues with the experimental design relating to attempts to link induced DNA damage to sleep regulation (outlined below). Moreover, although the methods used are appropriate for the study of the questions at hand, there are issues with the interpretation of the data and with these results being over-interpreted as evidence to support the paper's conclusions.

This study shows that a marker of DNA repair molecular machinery that is recruited to DNA double-strand breaks (γH2AX) is elevated in brain cells of the cavefish relative to the surface fish, and that reactive oxygen species are higher in most areas of the digestive tract of the cavefish than in that of the surface fish. As sleep deprivation has been previously linked to increases in both these parameters in other organisms (both vertebrates and invertebrates), their elevation in the cavefish morph is taken to indicated that the cavefish show signs of the physiological effects of chronic sleep deprivation.

It has been suggested that induction of DNA damage can directly drive sleep behaviour, with a notable study describing both the induction of DNA damage and an increase in sleep/immobility in zebrafish (*Danio rerio*) larvae by exposure to UV radiation (Zada et al. 2021 doi:10.1016/j.molcel.2021.10.026). In the present study, an increase in sleep/immobility is induced in surface fish larvae by exposure to UV light, but there is no effect on behaviour in cavefish larvae. This finding is interpreted as representing a loss of a sleep-promoting response to DNA damage in the cavefish morph. However, induction of DNA damage is not measured in this experiment, so it is not certain if similar levels of DNA damage are induced in each group of intact larvae, nor how the amount of damage induced compares to the pre-existing levels of DNA damage in the cavefish versus the surface fish larvae. In both this study with A. mexicanus surface morphs and the previous experiments from Zada et al. in zebrafish, observed increases in immobility following UV radiation exposure are interpreted as following from UV-induced DNA damage. However, in interpreting these experiments it is important to note that the cavefish morphs are eyeless and blind. Intense UV radiation is aversive to fish, and it has previously been shown in zebrafish larvae that (at least some) behavioural responses to UV exposure depend on the presence of an intact retina and UV-sensitive cone photoreceptors (Guggiana-Nilo and Engert, 2016, doi:10.3389/fnbeh.2016.00160). It is premature to conclude that the lack of behavioural response to UV exposure is in the cavefish is due to a difference response to DNA damage, as their lack of eyes will likely inhibit a response to the UV stimulus. Indeed, were the equivalent zebrafish experiment from Zada et al. to be repeated with mutant larvae fish lacking the retinal basis for UV detection it might be found that, in this case too, the effects of UV on behaviour are dependent on visual function. Such a finding should prompt a reappraisal of the interpretation that UV exposure's effects on fish sleep/locomotor behaviour are mediated by DNA damage. An additional note, relating to both Lloyd, Xia et al. and Zada et al., is that though increases in immobility are induced following UV exposure, in neither study have assays of sensory responsiveness been performed during this period. As a decrease in sensory responsiveness is a key behavioural criterion for defining sleep, it is therefore unclear that this post-UV behaviour is genuinely increased sleep as opposed to a stress-linked suppression of locomotion due to the intensely aversive UV stimulus. While it is true that behavioural immobility is used by many studies as a criterion to identify sleep in non-mammalian species, this is only fully appropriate when other elements of the behavioural criteria of sleep (e.g. reduced responsiveness to sensory stimuli, rapid reversibility, homeostatic regulation, circadian regulation) have been shown to be associated with these periods of behavioural quiescence. In both Lloyd, Xia et al. and Zada et al., only an increased immobility has been demonstrated, occurring at a period where the circadian clock would be promoting wake and natural homeostatic sleep drive would be expected to be at the low end of its normal range. At a minimum, testing sensory threshold would be advisable to ensure that the classification of this behaviour as sleep is accurate and to avoid the risk of being misled in the interpretation of these experiments.

The effects of UV exposure, in terms of causing damage to DNA, inducing DNA damage response and repair mechanisms, and in causing broader changes in gene expression are assessed in both surface and cavefish larvae, as well as in cell lines derived from these different morphs. Differences in the suite of DNA damage response mechanisms that are upregulated are shown to exist between surface fish and cavefish larvae, though at least some of this difference is likely to be due to differences gene expression that may exist even without UV exposure (this is discussed further below).

UV exposure induced DNA damage (as measured by levels of cyclobutene pyrimidine dimers) to a similar degree in cell lines derived from both surface fish and cave fish. However, γH2AX shows increased expression only in cells from the surface fish, suggesting an induction of an increased DNA repair response in these surface morphs, corroborated by their cells' increased ability to repair damaged DNA constructs experimentally introduced to the cells in a subsequent experiment. This "host cell reactivation assay" is a very interesting assay for measuring DNA repair in cell lines, but the power of this approach might be enhanced by introducing these DNA constructs into larval neurons in vivo (perhaps by electroporation) and by tracking DNA repair in living animals. Indeed, in such a preparation, the relationship between DNA repair and sleep/wake state could be assayed.

Comparing gene expression in tissues from young (here 1 year) and older (here 7-8 years) fish from both cavefish and surface fish morphs, the authors found that there are significant differences in the transcriptional profiles in brain and gut between young and old surface fish, but that for cavefish being 1 year old versus being 7-8 years old did not have a major effect on transcriptional profile. The authors take this as suggesting that there is a reduced transcriptional change occurring during aging and that the transcriptome of the cavefish is resistant to age-linked changes. This seems to be only one of the equally plausible interpretations of the results; it could also be the case that alterations in metabolic cellular and molecular mechanisms, and particularly in responses to DNA damage, in the cavefish mean that these fish adopt their "aged" transcriptome within the first year of life. This would mean that rather than the findings revealing that "the transcriptome of the cavefish is resilient to age-associated changes despite sleep loss, elevated ROS and elevated DNA damage", it would suggest that the cavefish transcriptome is sensitive to age-associated changes, potentially being driven by this low level of sleep, elevated reactive oxygen species, and elevated DNA damage. This alternative interpretation greatly changes the understanding of the present findings. One way in which the more correct interpretation could be determined would be by adding a further, younger group of fish to the comparison (perhaps a group in the age range of 1-3 months, relatively shortly after metamorphosis).

A major weakness of the study in its current form is the absence of sleep deprivation experiments to assay the effects of sleep loss on the cellular and molecular parameters in question. Without such experiments, the supposed link of sleep to the molecular, cellular and "aging" phenotypes remains tenuous. Although the argument might be made that the cavefish represent a naturally "sleep deprived" population, the cavefish in this study are not sleep deprived, rather they are adapted to a condition of reduced sleep relative to fish from surface populations. Comparing the effects of depriving fish from each morph on markers of DNA damage and repair, on gut reactive oxygen species, and on gene expression will be necessary to solidify any proposed link of these phenotypes to sleep.

A second important aspect that limits the interpretability and impact of this study is the absence of information about circadian variations in the parameters measured. A relationship between circadian phase, light exposure and DNA damage/repair mechanisms is known to exist in A. mexicanus and other teleosts, and for differences to exist between the cave and surface morphs in there phenomena (Beale et al. 2013, doi: 10.1038/ncomms3769). Although the present study mentions that their experiments do not align with these previous findings, they do not perform the appropriate experiments to determine if this such a misalignment is genuine. Specifically, Beale et al. 2013 showed that white light exposure drove enhanced expression of DNA repair genes (including cpdp which is prominent in the current study) in both surface fish and cavefish morphs, but that the magnitude of this change was less in the cave fish because they maintained an elevated expression of these genes in the dark, whereas darkness supressed the expression of these genes in the surface fish. If such a phenomenon is present in the setting of the current study, this would likely be a significant confound for the UV-induced gene expression experiments in intact larvae, and undermine the interpretation of the results derived from these experiments: as samples are collected 90 minutes after the dark-light transition (ZT 1.5) it would be expected that both cavefish and surface fish larvae should have a clear induction of DNA repair genes (including cpdp) regardless of 90s of UV exposure. The data in supplementary figure 3 is not sufficient to discount this potentially serious confound, as for larvae there is only gene expression data for timepoints from ZT2 to ZT 14, with all of these timepoints being in the light phase and not capturing any dynamics that would occur at the most important timepoints from ZT0-ZT1.5, in the relevant period after dark-light transition. Indeed, an appropriate control for this experiment would involve frequent sampling at least across 48 hours to assess light-linked and developmentally-related changes in gene expression that would occur in 5-6dpf larvae of each morph independently of the exposure to UV.

On a broader point, given the effects of both circadian rhythm and lighting conditions that are thought to exist in A. mexicanus (e.g. Beale et al. 2013) experiments involving measurements of DNA damage and repair, gene expression, and reactive oxygen species etc. at multiple times across >1 24 hour cycle, in both light-dark and constant illumination conditions (e.g. constant dark) would be needed to substantiate the authors' interpretation that their findings indicate consistently altered levels of these parameters in the cave fish relative to the surface fish. Most of the data in this study is taken at only single timepoints.

In summary, the authors show that there are differences in gene expression, activity of DNA damage response and repair pathways, response to UV radiation, and gut reactive oxygen species between the Pachón cavefish morph and the surface morph of Astyanax mexicanus. However, the data presented does not make the precise nature of these differences very clear, and the interpretation of the results appears to be overly strong. Furthermore, the evidence of a link between these morph specific differences and sleep is unconvincing.

Comments on revisions:

I thank the authors for their engagement with the notes and recommendations I made in my original comments. I have no further recommendations to make here.

---

## [Author Response]

The following is the authors’ response to the original reviews.

**Recommendations for the authors:**

**Reviewer #1 (Recommendations For The Authors):**
AbstractI don't think you need the first two sentences of the abstract. This is not a grant and your results are exciting enough to justify a full basic science-based approach.

We fully understand this perspective. However, we prefer to introduce the work in the broader context of sleep medicine. This manuscript is part of our long-standing efforts to develop cavefish as a model for sleep disorders and we believe this provides important context.

Last sentence of the abstract: the subject is missing. "That have developed..." who has developed?

Thank you. We have corrected this error, the sentence now reads “...these findings suggest that cavefish have developed resilience to sleep loss...”

IntroductionFirst paragraph. Worth explaining in a sentence what is the link between DNA damage and ROS.

We now state ‘Further, chronic sleep loss results in elevated reactive oxygen species (ROS), a known mediatior of DNA damage, in the gut and/or brain that contribute to mortality in *Drosophila* and mice [11,16].’

"A. mexicanus exists as blind cave populations and an extant surface population that are interfertile". This needs rephrasing. As it is, it sounds like the surface population is infertile.

We have rephrased for clarity; the line now reads: “while the surface and cave populations are geographically isolated, they remain interfertile and capable of hybridization in nature as well as laboratory settings”.

"Further, the evolved differences in DNA repair genes, including links between mechanisms regulating sleep, light responsiveness, and DNA repair across all three cave populations studied to date [27,29]" This sentence is incomplete.

We have corrected the phrasing, which now reads “...evolved differences in DNA repair genes have been identified across all three cave populations studied to date, including links between mechanisms regulating sleep, light responsiveness, and DNA repair”:

Figure 1I recommend improving the legibility of the figure copying some of the information provided in the legend directly within the figure itself.A, B: label in the panel itself what is blue and what is green.

Thank you, we have made this change.

C: Make it clear in the figure itself that you are measuring yH2AX. Also, probably you have enough room in the figure to avoid abbreviations for Rhomb, mes, and tele. It may also help if you could add a little cartoon that explains what those three brain regions are.

We have added text to the y axis indicating that yH2AX fluorescence is being measured, and replaced the abbreviations with eh full names of the regions.

G: again, explain that DHE is being measured here. And perhaps pick a different colour choice to highlight the difference from C?

We have added clarifiaction to the y-axis of the figure, but have retained the color scheme for consistency; in all surface-cave comparisons in the manuscript, gray is used for surface fish and red for cavefish.

In the text: I would recommend adding some quantitative reminder of what is the difference in sleep amount between the two species (cave vs surface).

We have added the following to highlight the magnitude of the difference in sleep: “Strikingly, cavefish sleep as little as 1-2 hours per day, in contrast to their surface counterparts, which sleep as much as 6-10 hours a day”

"Together, these findings fortify the notion that cellular stress is elevated in the gut of cavefish relative to surface fish." Were the two populations fed the same diet and raised in the same lab conditions? If this is pinpointed to sleep amount, it's worth ruling out possible confounding factors.

We have added a sentence to the results underlining this point: “Prior to imaging, both surface and cavefish had been reared in a temperature-controlled incubator, and relied solely on their yolk sac for nutrients; so, differences in gut ROS cannot be attributed to differences in rearing or feeding conditions.”

Figure 2Spell out, somewhere in the figure itself, that the 30s and 60s refer to UV treatment protocols.

We have added X-axis titles to clarify this in Fig 2 and supp. Fig 1.

It would be worth providing a cartoon of the experimental setup that shows for instance what time of the day UV was given (it's only specified in the text) and which subsequent sleep period was selected for comparisons.

We have added arrows to all sleep plots indicating the time of UV treatment, and brackets indicating the time period used for statistical comparisons, as well as text in the figure legends indicating this.

Figure 3A. I don't think this is needed, to be honest, and if you want to keep it, it needs a better legend.

We have edited the figure legend to increase clarity.

B. I would make it clear in the figure that this refers to transcriptomics analysis. Perhaps you could change the order and show C, D, and then B.

We have added text to the figure legend and the results text to more explicitly state that the PCA plot is of transcriptional response. We have however retained the original figure order, as well feel this figure is important to establish that both populations have strong, but distinct responses to the UV treatment.

Figure 4A. Spell it out in the figure itself that you're staining for CPD.

Thank you, we have made this change.

B. You are using the same colour combination you had in Figure 1 but for yet another pairing. This is a bit confusing.

Thank you for bringing this to our attention. We have added descriptions of the colors in the figure legend.

Discussion"Beyond the Pachón cavefish population, all three other cavefish populations have been found to have reduced sleep (Cite)." Citation missing here.

Thank you. We have now clarified this sentence and included a citation.

**Reviewer #2 (Recommendations For The Authors):**
Consideration of Environmental Conditions:Evaluate whether the lab conditions, which may more closely resemble surface environments, could influence the observed increase in neuronal DNA damage and gut ROS levels in cavefish. Adjusting these conditions or discussing their potential impact in the manuscript would strengthen the findings.

We are very excited about these experiments. We have a paper that will be submitted to *BioRxiv* this week where we record wild-caught fish, as well as fish in caves. The conclusion is that sleep loss is present in both populations. This field work took over 10 years to come together and still lacks the power of the lab based assays. Nevertheless, we can conclusively say that the phenotypes we have observed for the last ~15 years in the lab are present in a natural setting. We have included a statement about the need for future work to test these findings in a natural setting.

Alternative Stressors:Given that cavefish are albino and blind (to my knowledge), consider using alternative sources of genotoxic stress beyond UV-induced damage. This could include chemical agents or other forms of environmental stress to provide a more comprehensive assessment of DDR.

We agree and are enthusiastic about looking more generally at stress. We note that we have previously found that cavefish rebound following sleep deprivation (McGaugh et al, 2020) suggesting that they are responsive to sleep disruption. This will be a major research focus area moving forward.

Broader Stress Responses:Investigate whether other forms of stress, such as dietary changes or temperature fluctuations, elicit similar differences in sleep patterns and DDR responses. This could provide additional insights into the robustness of the observed phenomena.

We fully agree. This will be the primary focus of this research area moving forward. We hypothesize that cavefish are generally less responsive to their environment. Unpublished data reveals that temperature stress, circadian changes, and aging (presented here) to little to impact gene expression in surface fish. We would like to test the hypothesis that transcriptional stability of cavefish contributes to their longevity.

Potential Protective Mechanisms:Discuss the possibility that lower levels of gamma-H2AX in cavefish might be protective, as DDR can lead to cellular senescence or cancer. This perspective could add depth to the interpretation of the results.

This was the hypothesis underlying this manuscript. However, we found elevated levels of gamma-H2AX. We believe there may be additional protective mechanisms that have evolved in cavefish, but cannot identify them to date. Our hope is future functional studies by our group, as well as other groups’ access to this published work, may help address these questions.

Strengthening the Sleep-DNA Damage Link:Further experiments are needed to directly link sleep differences to the observed variations in DNA damage and DDR. This could involve manipulating sleep patterns in surface fish and cavefish to observe corresponding changes in DNA repair mechanisms.

We agree. We have referenced work that conclusively showed this relationship in zebrafish. Our current methods for limiting sleep involves shaking, and this has too many confounds. We are working on developing genetic tools, and applying the gentle rocking methods used previously in zebrafish to address these questions.

Clarification of Causal Directionality:Address the potential that sleep patterns and DDR responses may both be downstream effects of a common cause or independent adaptations to the cave environment. Clarifying this in the manuscript would provide a more nuanced understanding of the evolutionary adaptations.

Thank you for this suggestion. We have now added a paragraph describing how these experiments (and the ones described above) are necessary for understanding the relationship between sleep and DDR.

Clarification and Presentation:Fix the many typos, and improve the clarity of the figures and their legends to ensure they are easily interpretable. Additional context in the discussion section would help readers understand the significance and potential implications of the findings.

Thank you, we have now included this.

**Reviewer #3 (Recommendations For The Authors):**
There are a number of suggestions that I have made in the public review, but there are a few things that I would like to add here.The methods section is missing many important details, for instance, the intensity of the illumination used in the UV exposure in larvae is not reported but is vital for the interpretation/replication of these experiments. In general, this section should be redone with a greater effort to include all important information. Similarly, the figure legends could be greatly improved, with important details like n-number and definition of significance thresholds defined (e.g. see Figures 1, C, and G.)

We have added greater detail to the methods section to specify the spectral peak and power output of the bulbs used.

There are a number of passages in the manuscript that do not make sense, which suggests that a future version of record should be carefully proofread. I know that this can be a case of reading multiple versions of a manuscript so many times that one doesn't really see it anymore, but, for example, phrases like "To differentiate between these two possibilities" are confusing to the reader when there has been no introduction of alternate possibilities.

Thank you for this comment. We have fixed this mistake and proofread the manuscript.

Additionally, there are multiple examples of errors in citations/references. A few examples are below:"Further, chronic sleep loss results in elevated reactive oxygen species (ROS) in the gut and/or brain that contribute to mortality in Drosophila and mice [11, 16]". Reference 16 does not include mice at all, and reference 11 is Vaccaro et al. 2020, where Drosophila mortality is assessed, but mouse mortality is not.

We have added the appropriate citations and revised this sentence.

References 13 and 15 are the same.

Thank you, we have fixed.

References 24 and 26 are the same.

Thank you, we have fixed.

**Public Reviews:**

**Reviewer #1 (Publc Review):**
Summary:Lloyd et al employ an evolutionary comparative approach to study how sleep deprivation affects DNA damage repair in Astyanax mexicanus, using the cave vs surface species evolution as a playground. The work shows, convincingly, that the cavefish population has evolved an impaired DNA damage response both following sleep deprivation or a classical paradigm of DNA damage (UV).Strengths:The study employs a thorough multidisciplinary approach. The experiments are well conducted and generally well presented.Weaknesses:Having a second experimental mean to induce DNA damage would strengthen and generalise the findings.Overall, the study represents a very important addition to the field. The model employed underlines once more the importance of using an evolutionary approach to study sleep and provides context and caveats to statements that perhaps were taken a bit too much for granted before. At the same time, the paper manages to have an extremely constructive approach, presenting the platform as a clear useful tool to explore the molecular aspects behind sleep and cellular damage in general. The discussion is fair, highlighting the strengths and weaknesses of the work and its implications.

We fully agree with this assessment. We are currently performing experiments to test the effects of additional DNA damaging agents. We hope to extend these studies beyond DNA-damage agents to look more generally at how animals respond to stress including ROS, sleep deprivation, and high temperature. This will be a major direction of the laboratory moving forward.

The manuscript investigates the relationship between sleep, DNA damage, and aging in the Mexican cavefish (Astyanax mexicanus), a species that exhibits significant differences in sleep patterns between surface-dwelling and cave-dwelling populations. The authors aim to understand whether these evolved sleep differences influence the DNA damage response (DDR) and oxidative stress levels in the brain and gut of the fish.Summary of the Study:The primary objective of the study is to determine if the reduced sleep observed in cave-dwelling populations is associated with increased DNA damage and altered DDR. The authors compared levels of DNA damage markers and oxidative stress in the brains and guts of surface and cavefish. They also analyzed the transcriptional response to UV-induced DNA damage and evaluated the DDR in embryonic fibroblast cell lines derived from both populations.Strengths of the Study:Comparative Approach:The study leverages the unique evolutionary divergence between surface and cave populations of A. mexicanus to explore fundamental biological questions about sleep and DNA repair.Multifaceted Methodology:The authors employ a variety of methods, including immunohistochemistry, RNA sequencing, and in vitro cell line experiments, providing a comprehensive examination of DDR and oxidative stress.Interesting Findings:The study presents intriguing results showing elevated DNA damage markers in cavefish brains and increased oxidative stress in cavefish guts, alongside a reduced transcriptional response to UV-induced DNA damage.Weaknesses of the Study:Link to Sleep Physiology:The evidence connecting the observed differences in DNA damage and DDR directly to sleep physiology is not convincingly established. While the study shows distinct DDR patterns, it does not robustly demonstrate that these are a direct result of sleep differences.

We agree with this assessment. We are currently working to apply tools developed in zebrafish to examine the physiology of sleep. While this is important, and our results our promising, we will note that functional analysis of sleep physiology in fish has been limited to zebrafish. We hope future studies will allow us to integrate approaches that examine the physiology of sleep.

Causal Directionality:The study fails to establish a clear causal relationship between sleep and DNA damage. It is possible that both sleep patterns and DDR responses are downstream effects of a common cause or independent adaptations to the cave environment.

We agree, however, we note that this could be the case for all animals in which sleep has been linked to DNA damage. We believe the most likely explanation for *Astyanax* and other animals studied, is that sleep is that sleep and DDR are downstream/interface with the sleep homeostat.

Environmental Considerations:The lab conditions may not fully replicate the natural environments of the cavefish, potentially influencing the results. The impact of these conditions on the study's findings needs further consideration.

This is correct. We have considered this carefully. After nearly a decade of effort, we have completed analysis of sleep in the wild. These will be uploaded to BioRxiv within the next week.

Photoreactivity in Albino Fish:The use of UV-induced DNA damage as a primary stressor may not be entirely appropriate for albino, blind cavefish. Alternative sources of genotoxic stress should be explored to validate the findings.

We have addressed this above. Future work will examine additional stressors. Both fish are transparent at 6dpf and so it is unlikely that albinism impacts the amount of UV that reaches the brain.

Assessment of the Study's Achievements:The authors partially achieve their aims by demonstrating differences in DNA damage and DDR between surface and cavefish. However, the results do not conclusively support the claim that these differences are driven by or directly related to the evolved sleep patterns in cavefish. The study's primary claims are only partially supported by the data.Impact and Utility:The findings contribute valuable insights into the relationship between sleep and DNA repair mechanisms, highlighting potential areas of resilience to DNA damage in cavefish. While the direct link to sleep physiology remains unsubstantiated, the study's data and methods will be useful to researchers investigating evolutionary biology, stress resilience, and the molecular basis of sleep.
**Reviewer #3 (Public Review):**
Lloyd, Xia, et al. utilised the existence of surface-dwelling and cave-dwelling morphs of Astyanax mexicanus to explore a proposed link between DNA damage, aging, and the evolution of sleep. Key to this exploration is the behavioural and physiological differences between cavefish and surface fish, with cavefish having been previously shown to have low levels of sleep behaviour, along with metabolic alterations (for example chronically elevated blood glucose levels) in comparison to fish from surface populations. Sleep deprivation, metabolic dysfunction, and DNA damage are thought to be linked and to contribute to aging processes. Given that cavefish seem to show no apparent health consequences of low sleep levels, the authors suggest that they have evolved resilience to sleep loss. Furthermore, as extended wake and loss of sleep are associated with increased rates of damage to DNA (mainly double-strand breaks) and sleep is linked to repair of damaged DNA, the authors propose that changes in DNA damage and repair might underlie the reduced need for sleep in the cavefish morphs relative to their surface-dwelling conspecifics.To fulfill their aim of exploring links between DNA damage, aging, and the evolution of sleep, the authors employ methods that are largely appropriate, and comparison of cavefish and surface fish morphs from the same species certainly provides a lens by which cellular, physiological and behavioural adaptations can be interrogated. Fluorescence and immunofluorescence are used to measure gut reactive oxygen species and markers of DNA damage and repair processes in the different fish morphs, and measurements of gene expression and protein levels are appropriately used. However, although the sleep tracking and quantification employed are quite well established, issues with the experimental design relate to attempts to link induced DNA damage to sleep regulation (outlined below). Moreover, although the methods used are appropriate for the study of the questions at hand, there are issues with the interpretation of the data and with these results being over-interpreted as evidence to support the paper's conclusions.This study shows that a marker of DNA repair molecular machinery that is recruited to DNA double-strand breaks (γH2AX) is elevated in brain cells of the cavefish relative to the surface fish and that reactive oxygen species are higher in most areas of the digestive tract of the cavefish than in that of the surface fish. As sleep deprivation has been previously linked to increases in both these parameters in other organisms (both vertebrates and invertebrates), their elevation in the cavefish morph is taken to indicate that the cavefish show signs of the physiological effects of chronic sleep deprivation.It has been suggested that induction of DNA damage can directly drive sleep behaviour, with a notable study describing both the induction of DNA damage and an increase in sleep/immobility in zebrafish (*Danio rerio*) larvae by exposure to UV radiation (Zada et al. 2021 doi:10.1016/j.molcel.2021.10.026). In the present study, an increase in sleep/immobility is induced in surface fish larvae by exposure to UV light, but there is no effect on behaviour in cavefish larvae. This finding is interpreted as representing a loss of a sleep-promoting response to DNA damage in the cavefish morph. However, induction of DNA damage is not measured in this experiment, so it is not certain if similar levels of DNA damage are induced in each group of intact larvae, nor how the amount of damage induced compares to the pre-existing levels of DNA damage in the cavefish versus the surface fish larvae. In both this study with A. mexicanus surface morphs and the previous experiments from Zada et al. in zebrafish, observed increases in immobility following UV radiation exposure are interpreted as following from UV-induced DNA damage. However, in interpreting these experiments it is important to note that the cavefish morphs are eyeless and blind. Intense UV radiation is aversive to fish, and it has previously been shown in zebrafish larvae that (at least some) behavioural responses to UV exposure depend on the presence of an intact retina and UV-sensitive cone photoreceptors (Guggiana-Nilo and Engert, 2016, doi:10.3389/fnbeh.2016.00160). It is premature to conclude that the lack of behavioural response to UV exposure in the cavefish is due to a different response to DNA damage, as their lack of eyes will likely inhibit a response to the UV stimulus.

We believe that in *A. mexicanus,* like in zebrafish, it is highly unlikely that the effects of UV are mediated through visual processing. Even if this were the case, the timeframe of UV activation is very short compared to the time-scale of sleep measurements so this is unlikely to be a confound.

Indeed, were the equivalent zebrafish experiment from Zada et al. to be repeated with mutant larvae fish lacking the retinal basis for UV detection it might be found that in this case too, the effects of UV on behaviour are dependent on visual function. Such a finding should prompt a reappraisal of the interpretation that UV exposure's effects on fish sleep/locomotor behaviour are mediated by DNA damage.

We prefer not to comment on Zada et al, as that is a separate manuscript.

An additional note, relating to both Lloyd, Xia, et al., and Zada et al., is that though increases in immobility are induced following UV exposure, in neither study have assays of sensory responsiveness been performed during this period. As a decrease in sensory responsiveness is a key behavioural criterion for defining sleep, it is, therefore, unclear that this post-UV behaviour is genuinely increased sleep as opposed to a stress-linked suppression of locomotion due to the intensely aversive UV stimulus.

We understand this concern and are working on improved methodology for measuring sleep. However, behavioral measurements are the standard for almost every manuscript that has studied sleep in zebrafish, flies, and worms to date.

The effects of UV exposure, in terms of causing damage to DNA, inducing DNA damage response and repair mechanisms, and in causing broader changes in gene expression are assessed in both surface and cavefish larvae, as well as in cell lines derived from these different morphs. Differences in the suite of DNA damage response mechanisms that are upregulated are shown to exist between surface fish and cavefish larvae, though at least some of this difference is likely to be due to differences in gene expression that may exist even without UV exposure (this is discussed further below).UV exposure induced DNA damage (as measured by levels of cyclobutene pyrimidine dimers) to a similar degree in cell lines derived from both surface fish and cave fish. However, γH2AX shows increased expression only in cells from the surface fish, suggesting induction of an increased DNA repair response in these surface morphs, corroborated by their cells' increased ability to repair damaged DNA constructs experimentally introduced to the cells in a subsequent experiment. This "host cell reactivation assay" is a very interesting assay for measuring DNA repair in cell lines, but the power of this approach might be enhanced by introducing these DNA constructs into larval neurons in vivo (perhaps by electroporation) and by tracking DNA repair in living animals. Indeed, in such a preparation, the relationship between DNA repair and sleep/wake state could be assayed.Comparing gene expression in tissues from young (here 1 year) and older (here 7-8 years) fish from both cavefish and surface fish morphs, the authors found that there are significant differences in the transcriptional profiles in brain and gut between young and old surface fish, but that for cavefish being 1 year old versus being 7-8 years old did not have a major effect on transcriptional profile. The authors take this as suggesting that there is a reduced transcriptional change occurring during aging and that the transcriptome of the cavefish is resistant to age-linked changes. This seems to be only one of the equally plausible interpretations of the results; it could also be the case that alterations in metabolic cellular and molecular mechanisms, and particularly in responses to DNA damage, in the cavefish mean that these fish adopt their "aged" transcriptome within the first year of life.

This is indeed true. However, one could also interpret this as a lack of aging. If the profile does not change over time, the difference seems largely semantic.

A major weakness of the study in its current form is the absence of sleep deprivation experiments to assay the effects of sleep loss on the cellular and molecular parameters in question. Without such experiments, the supposed link of sleep to the molecular, cellular, and "aging" phenotypes remains tenuous. Although the argument might be made that the cavefish represent a naturally "sleep-deprived" population, the cavefish in this study are not sleep-deprived, rather they are adapted to a condition of reduced sleep relative to fish from surface populations. Comparing the effects of depriving fish from each morph on markers of DNA damage and repair, gut reactive oxygen species, and gene expression will be necessary to solidify any proposed link of these phenotypes to sleep.

We agree this would be beneficial. We note that relatively few papers have sleep deprived fish. While we done have this before in A. mexicanus the assay is less than ideal and likely induces generalizable stress. We are working on adapting more recently developed methods in zebrafish.

A second important aspect that limits the interpretability and impact of this study is the absence of information about circadian variations in the parameters measured. A relationship between circadian phase, light exposure, and DNA damage/repair mechanisms is known to exist in A. mexicanus and other teleosts, and differences exist between the cave and surface morphs in their phenomena (Beale et al. 2013, doi: 10.1038/ncomms3769). Although the present study mentions that their experiments do not align with these previous findings, they do not perform the appropriate experiments to determine if such a misalignment is genuine. Specifically, Beale et al. 2013 showed that white light exposure drove enhanced expression of DNA repair genes (including cpdp which is prominent in the current study) in both surface fish and cavefish morphs, but that the magnitude of this change was less in the cave fish because they maintained an elevated expression of these genes in the dark, whereas the darkness suppressed the expression of these genes in the surface fish. If such a phenomenon is present in the setting of the current study, this would likely be a significant confound for the UV-induced gene expression experiments in intact larvae, and undermine the interpretation of the results derived from these experiments: as samples are collected 90 minutes after the dark-light transition (ZT 1.5) it would be expected that both cavefish and surface fish larvae should have a clear induction of DNA repair genes (including cpdp) regardless of 90s of UV exposure. The data in Supplementary Figure 3 is not sufficient to discount this potentially serious confound, as for larvae there is only gene expression data for time points from ZT2 to ZT 14, with all of these time points being in the light phase and not capturing any dynamics that would occur at the most important timepoints from ZT0-ZT1.5, in the relevant period after dark-light transition. Indeed, an appropriate control for this experiment would involve frequent sampling at least across 48 hours to assess light-linked and developmentally-related changes in gene expression that would occur in 5-6dpf larvae of each morph independently of the exposure to UV.

We agree that this would be useful, however, frequent sampling is not feasible given the experiments presented here and the challenges of working with an emerging model.

On a broader point, given the effects of both circadian rhythm and lighting conditions that are thought to exist in A. mexicanus (e.g. Beale et al. 2013) experiments involving measurements of DNA damage and repair, gene expression, and reactive oxygen species, etc. at multiple times across >1 24 hour cycle, in both light-dark and constant illumination conditions (e.g. constant dark) would be needed to substantiate the authors' interpretation that their findings indicate consistently altered levels of these parameters in the cavefish relative to the surface fish. Most of the data in this study is taken at only single time points.

Again, see comment above. The goal was to identify whether there are differences in DNA Damage response between *A. mexcicanus.* Extending on this to examine interactions with the circadian system could be a useful path to pursue in the future.

On a broader point, given the effects of both circadian rhythm and lighting conditions that are thought to exist in A. mexicanus (e.g. Beale et al. 2013) experiments involving measurements of DNA damage and repair, gene expression, and reactive oxygen species, etc. at multiple times across >1 24 hour cycle, in both light-dark and constant illumination conditions (e.g. constant dark) would be needed to substantiate the authors' interpretation that their findings indicate consistently altered levels of these parameters in the cavefish relative to the surface fish. Most of the data in this study is taken at only single time points.In summary, the authors show that there are differences in gene expression, activity of DNA damage response and repair pathways, response to UV radiation, and gut reactive oxygen species between the Pachón cavefish morph and the surface morph of Astyanax mexicanus. However, the data presented does not make the precise nature of these differences very clear, and the interpretation of the results appears to be overly strong. Furthermore, the evidence of a link between these morph-specific differences and sleep is unconvincing.In summary, the authors show that there are differences in gene expression, activity of DNA damage response and repair pathways, response to UV radiation, and gut reactive oxygen species between the Pachón cavefish morph and the surface morph of Astyanax mexicanus. However, the data presented does not make the precise nature of these differences very clear, and the interpretation of the results appears to be overly strong. Furthermore, the evidence of a link between these morph-specific differences and sleep is unconvincing.